# F5-TTS: A Fairytaler that Fakes Fluent and Faithful Speech with Flow Matching

## Abstract

This paper introduces F5-TTS, a fully non-autoregressive text-to-speech system based on flow matching with Diffusion Transformer (DiT). Without requiring complex designs such as duration model, text encoder, and phoneme alignment, the text input is simply padded with filler tokens to the same length as input speech, and then the denoising is performed for speech generation, which was originally proved feasible by E2 TTS. However, the original design of E2 TTS makes it hard to follow due to its slow convergence and low robustness. To address these issues, we first model the input with ConvNeXt to refine the text representation, making it easy to align with the speech. We further propose an inference-time Sway Sampling strategy, which significantly improves our model's performance and efficiency. This sampling strategy for flow step can be easily applied to existing flow matching based models without retraining. Our design allows faster training and achieves an inference RTF of 0.15, which is greatly improved compared to state-of-the-art diffusion-based TTS models. Trained on a public 100K hours multilingual dataset, our Fairytaler Fakes Fluent and Faithful speech with Flow matching (F5-TTS) exhibits highly natural and expressive zero-shot ability, seamless code-switching capability, and speed control efficiency. Demo samples can be found at `https://F5-TTS.github.io`. We will release all code and checkpoints to promote community development.

## 1 Introduction

Recent research in Text-to-Speech (TTS) has experienced great advancement (Shen et al., 2018; Li et al., 2019; Ren et al., 2020; Kim et al., 2020; 2021; Popov et al., 2021; Wang et al., 2023b; Tan et al., 2024). With a few seconds of audio prompt, current TTS models are able to synthesize speech for any given text and mimic the speaker of audio prompt (Wang et al., 2023a; Zhang et al., 2023b). The synthesized speech can achieve high fidelity and naturalness that they are almost indistinguishable from human speech (Shen et al., 2023; Ju et al., 2024; Chen et al., 2024; Le et al., 2024).

While autoregressive (AR) based TTS models exhibit an intuitive way of consecutively predicting the next token(s) and have achieved promising zero-shot TTS capability, the inherent limitations of AR modeling require extra efforts addressing issues such as inference latency and exposure bias (Song et al., 2024; Du et al., 2024a; Han et al., 2024; Xin et al., 2024; Peng et al., 2024). Moreover, the quality of speech tokenizer is essential for AR models to achieve high-fidelity synthesis (Zeghidour et al., 2021; Défossez et al., 2022; Wu et al., 2023; Yang et al., 2023; Zhang et al., 2023a; Bai et al., 2024; Niu et al., 2024). Thus, there have been studies exploring direct modeling in continuous space (Liu et al., 2024a; Li et al., 2024a; Meng et al., 2024) to enhance synthesized speech quality recently.

Although AR models demonstrate impressive zero-shot performance as they perform implicit duration modeling and can leverage diverse sampling strategies, non-autoregressive (NAR) models benefit from fast inference through parallel processing, and effectively balance synthesis quality and latency. Notably, diffusion models (Ho et al., 2020; Song et al., 2020) contribute most to the success of current NAR speech models (Shen et al., 2023; Ju et al., 2024). In particular, Flow Matching with Optimal Transport path (FM-OT) (Lipman et al., 2022) is widely used in recent research fields not only text-to-speech (Le et al., 2024; Guo et al., 2024b; Mehta et al., 2024; Lee et al., 2024; Eskimez et al., 2024) but also image generation (Esser et al., 2024) and music generation (Fei et al., 2024).

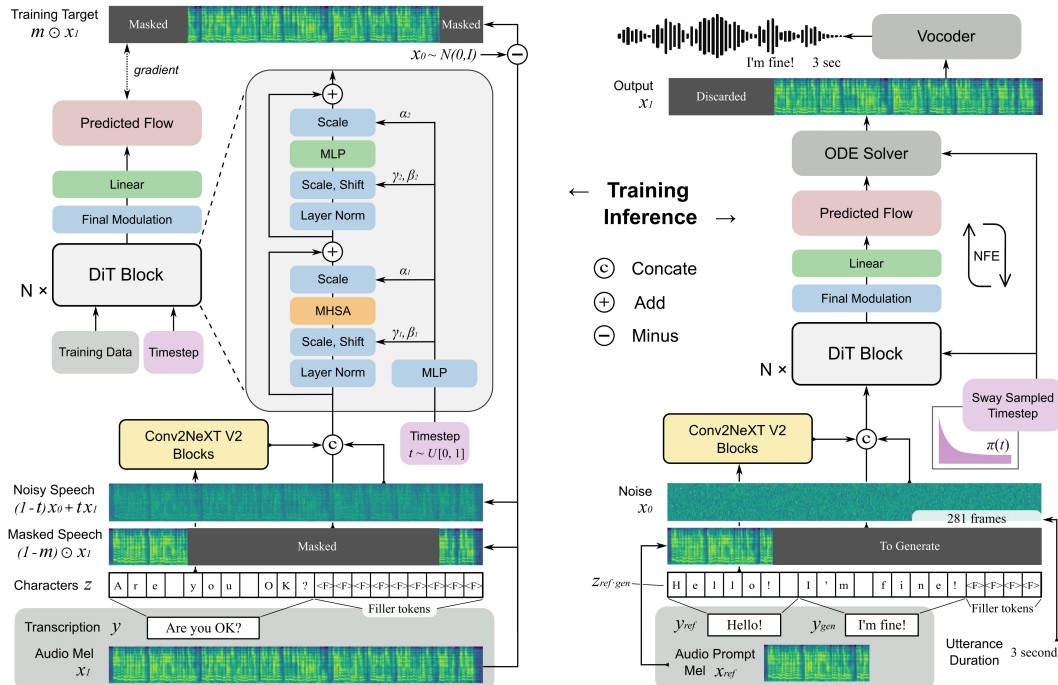

Figure 1: An overview of F5-TTS training (left) and inference (right). The model is trained on the text-guided speech-infilling task and condition flow matching loss. The input text is converted to a character sequence, padded with filler tokens to the same length as input speech, and refined by ConvNeXt blocks before concatenation with speech input. The inference leverages Sway Sampling for flow steps, with the model and an ODE solver to generate speech from sampled noise.

Unlike AR-based models, the alignment modeling between input text and synthesized speech is crucial and challenging for NAR-based models. While NaturalSpeech 3 (Ju et al., 2024) and Voicebox (Le et al., 2024) use frame-wise phoneme alignment; Matcha-TTS (Mehta et al., 2024) adopts monotonic alignment search (Kim et al., 2020) and relies on a phoneme-level duration model; recent works find that introducing such rigid alignment between text and speech hinders the model from generating results with higher naturalness (Eskimez et al., 2024; Anastassiou et al., 2024).

E3 TTS (Gao et al., 2023a) abandons phoneme-level duration and applies cross-attention on the input sequence but yields limited audio quality. DiTTo-TTS (Lee et al., 2024) uses Diffusion Transformer (DiT) (Peebles & Xie, 2023) with cross-attention conditioned on encoded text from a pretrained language model. To further enhance alignment, it uses the pretrained language model to finetune the neural audio codec, infusing semantic information into the generated representations. In contrast, E2 TTS (Eskimez et al., 2024), based on Voicebox (Le et al., 2024), adopts a simpler way, which removes the phoneme and duration predictor and directly uses characters padded with filler tokens to the length of mel spectrograms as input. This simple scheme also achieves very natural and realistic synthesized results. However, we found that robustness issues exist in E2 TTS for the text and speech alignment. Seed-TTS (Anastassiou et al., 2024) employs a similar strategy and achieves excellent results, though not elaborated in model details. In these ways of not explicitly modeling phoneme-level duration, models learn to assign the length of each word or phoneme according to the given total sequence length, resulting in improved prosody and rhythm.

In this paper, we propose **F5-TTS**, a **F**airytaler that **F**akes **F**luent and **F**aithful speech with **F**low matching. Maintaining the simplicity of pipeline without phoneme alignment, duration predictor, text encoder, and semantically infused codec model, F5-TTS leverages the Diffusion Transformer with ConvNeXt V2 (Woo et al., 2023) to better tackle text-speech alignment during in-context learning. We stress the deep entanglement of semantic and acoustic features in the E2 TTS model design, which has inherent problems and will pose alignment failure issues that could not simply be solved with re-ranking. With in-depth ablation studies, our proposed F5-TTS demonstrates stronger robust-

ness, in generating more faithful speech to the text prompt, while maintaining comparable speaker similarity. Additionally, we introduce an inference-time sampling strategy for flow steps substantially improving naturalness, intelligibility, and speaker similarity of generation. This approach can be seamlessly integrated into existing flow matching based models without retraining.

## 2 PRELIMINARIES

### 2.1 FLOW MATCHING

The Flow Matching (FM) objective is to match a probability path $p_t$ from a simple distribution $p_0$, $e.g.$, the standard normal distribution $p(x) = \mathcal{N}(x|0, I)$, to $p_1$ approximating the data distribution $q$. In short, the FM loss regresses the vector field $u_t$ with a neural network $v_t$ as

$$\mathcal{L}_{FM}(\theta) = E_{t,p_t(x)} \|v_t(x) - u_t(x)\|^2, \tag{1}$$

where $\theta$ parameterizes the neural network, $t \sim \mathcal{U}[0, 1]$ and $x \sim p_t(x)$. The model $v_t$ is trained over the entire flow step and data range, ensuring it learns to handle the entire transformation process from the initial distribution to the target distribution.

As we have no prior knowledge of how to approximate $p_t$ and $u_t$, a conditional probability path $p_t(x|x_1) = \mathcal{N}(x \mid \mu_t(x_1), \sigma_t(x_1)^2 I)$ is considered in actual training, and the Conditional Flow Matching (CFM) loss is proved to have identical gradients $w.r.t.$ $\theta$ (Lipman et al., 2022). $x_1$ is the random variable corresponding to training data. $\mu$ and $\sigma$ is the time-dependent mean and scalar standard deviation of Gaussian distribution.

Remember that the goal is to construct target distribution (data samples) from initial simple distribution, $e.g.$, Gaussian noise. With the conditional form, the flow map $\psi_t(x) = \sigma_t(x_1)x + \mu_t(x_1)$ with $\mu_0(x_1) = 0$ and $\sigma_0(x_1) = 1$, $\mu_1(x_1) = x_1$ and $\sigma_1(x_1) = 0$ is made to have all conditional probability paths converging to $p_0$ and $p_1$ at the start and end. The flow thus provides a vector field $d\psi_t(x_0)/dt = u_t(\psi_t(x_0)|x_1)$. Reparameterize $p_t(x|x_1)$ with $x_0$, we have

$$\mathcal{L}_{\text{CFM}}(\theta) = E_{t,q(x_1),p(x_0)} \|v_t(\psi_t(x_0)) - \frac{d}{dt}\psi_t(x_0)\|^2. \tag{2}$$

Further leveraging Optimal Transport form $\psi_t(x) = (1 - t)x + tx_1$, we have the OT-CFM loss,

$$\mathcal{L}_{\text{CFM}}(\theta) = E_{t,q(x_1),p(x_0)} \|v_t((1 - t)x_0 + tx_1) - (x_1 - x_0)\|^2. \tag{3}$$

To view in a more general way (Kingma & Gao, 2024), if formulating the loss in terms of log signal-to-noise ratio (log-SNR) $\lambda$ instead of flow step $t$, and parameterizing to predict $x_0$ ($\epsilon$, commonly stated in diffusion model) instead of predict $x_1 - x_0$, the CFM loss is equivalent to the v-prediction (Salimans & Ho, 2022) loss with cosine schedule.

For inference, given sampled noise $x_0$ from initial distribution $p_0$, flow step $t \in [0, 1]$ and condition with respect to generation task, the ordinary differential equation (ODE) solver (Chen, 2018) is used to evaluate $\psi_1(x_0)$ the integration of $d\psi_t(x_0)/dt$ with $\psi_0(x_0) = x_0$. The number of function evaluations (NFE) is the times going through the neural network as we may provide multiple flow step values from 0 to 1 as input to approximate the integration. Higher NFE will produce more accurate results and certainly take more calculation time.

### 2.2 CLASSIFIER-FREE GUIDANCE

Classifier Guidance (CG) is proposed by Dhariwal & Nichol (2021), functions by adding the gradient of an additional classifier, while such an explicit way to condition the generation process may have several problems. Extra training of the classifier is required and the generation result is directly affected by the quality of the classifier. Adversarial attacks might also occur as the guidance is introduced through the way of updating the gradient. Thus deceptive images with imperceptible details to human eyes may be generated, which are not conditional.

Classifier-Free Guidance (CFG) (Ho & Salimans, 2022) proposes to replace the explicit classifier with an implicit classifier without directly computing the explicit classifier and its gradient. The gradient of a classifier can be expressed as a combination of conditional generation probability and

unconditional generation probability. By dropping the condition with a certain rate during training, and linear extrapolating the inference outputs with and without condition $c$, the final guided result is obtained. We could balance between fidelity and diversity of the generated samples with

$$v_{t,CFG} = v_t(\psi_t(x_0), c) + \alpha(v_t(\psi_t(x_0), c) - v_t(\psi_t(x_0))) \tag{4}$$

in CFM case, where $\alpha$ is the CFG strength.[1]

## 3 METHOD

This work aims to build a high-level text-to-speech synthesis system. We trained our model on the text-guided speech-infilling task (Bai et al., 2022; Le et al., 2024). Based on recent research (Lee et al., 2024; Eskimez et al., 2024; Liu et al., 2024b), it is promising to train without phoneme-level duration predictor and can achieve higher naturalness in zero-shot generation deprecating explicit phoneme-level alignment. We adopt a similar pipeline as E2 TTS (Eskimez et al., 2024) and propose our advanced architecture F5-TTS, addressing the slow convergence (timbre learned well at an early stage but struggled to learn alignment) and robustness issues (failures on hard case generation) of E2 TTS. We also propose a Sway Sampling strategy for flow steps at inference, which significantly improves our model's performance in faithfulness to reference text and speaker similarity.

### 3.1 PIPELINE

**Training** The infilling task is to predict a segment of speech given its surrounding audio and full text (for both surrounding transcription and the part to generate). For simplicity, we reuse the symbol $x$ to denote an audio sample and $y$ the corresponding transcript for a data pair $(x, y)$. As shown in Fig.1 (left), the acoustic input for training is an extracted mel spectrogram features $x_1 \in \mathbb{R}^{F \times N}$ from the audio sample $x$, where $F$ is mel dimension and $N$ is the sequence length. In the scope of CFM, we pass in the model the noisy speech $(1 - t)x_0 + tx_1$ and the masked speech $(1 - m) \odot x_1$, where $x_0$ denotes sampled Gaussian noise, $t$ is sampled flow step, and $m \in \{0, 1\}^{F \times N}$ represents a binary temporal mask.

Following E2 TTS, we directly use alphabets and symbols for English. We opt for full pinyin to facilitate Chinese zero-shot generation. By breaking the raw text into such character sequence and padding it with filler tokens $\langle F \rangle$ to the same length as mel frames, we form an extended sequence $z$ with $c_i$ denoting the $i$-th character:

$$z = (c_1, c_2, \ldots, c_M, \underbrace{\langle F \rangle, \ldots, \langle F \rangle}_{(N-M) \text{ times}}). \tag{5}$$

The model is trained to reconstruct $m \odot x_1$ with $(1 - m) \odot x_1$ and $z$, which equals to learn the target distribution $p_1$ in form of $P(m \odot x_1 | (1 - m) \odot x_1, z)$ approximating real data distribution $q$.

**Inference** To generate a speech with the desired content, we have the audio prompt's mel spectrogram features $x_{ref}$, its transcription $y_{ref}$, and a text prompt $y_{gen}$. Audio prompt serves to provide speaker characteristics and text prompt is to guide the content of generated speech.

The sequence length $N$, or duration, has now become a pivotal factor that necessitates informing the model of the desired length for sample generation. One could train a separate model to predict and deliver the duration based on $x_{ref}$, $y_{ref}$ and $y_{gen}$. Here we simply estimate the duration based on the ratio of the number of characters in $y_{gen}$ and $y_{ref}$. We assume that the sum-up length of characters is no longer than mel length, thus padding with filler tokens is done as during training.

To sample from the learned distribution, the converted mel features $x_{ref}$, along with concatenated and extended character sequence $z_{ref \cdot gen}$ serve as the condition in Eq.4. We have

$$v_t(\psi_t(x_0), c) = v_t((1 - t)x_0 + tx_1 | x_{ref}, z_{ref \cdot gen}), \tag{6}$$

See from Fig.1 (right), we start from a sampled noise $x_0$, and what we want is the other end of flow $x_1$. Thus we use the ODE solver to gradually integrate from $\psi_0(x_0) = x_0$ to $\psi_1(x_0) = x_1$,

---

[1]Note that the inference time will be doubled if CFG. Model $v_t$ will execute the forward process twice, once with condition, and once without.

given $d\psi_t(x_0)/dt = v_t(\psi_t(x_0), x_{ref}, z_{ref\cdot gen})$. During inference, the flow steps are provided in an ordered way, *e.g.*, uniformly sampled a certain number from 0 to 1 according to the NFE setting.

After getting the generated mel with model $v_t$ and ODE solver, we discard the part of $x_{ref}$. Then we leverage a vocoder to convert the mel back to speech signal.

## 3.2   F5-TTS

E2 TTS directly concatenates the padded character sequence with input speech, thus deeply entangling semantic and acoustic features with a large length gap of effective information, which is the underlying cause of hard training and poses several problems in a zero-shot scenario (Sec.5.1). To alleviate the problem of slow convergence and low robustness, we propose F5-TTS which accelerates training and inference and shows a strong robustness in generation. Also, an inference-time Sway Sampling is introduced, which allows inference faster (using less NFE) while maintaining performance. This sampling way of flow step can be directly applied to other CFM models.

**Model**   As shown in Fig.1, we use latent Diffusion Transformer (DiT) (Peebles & Xie, 2023) as backbone. To be specific, we use DiT blocks with zero-initialized adaptive Layer Norm (adaLN-zero). To enhance the model's alignment ability, we also leverage ConvNeXt V2 blocks (Woo et al., 2023). Its predecessor ConvNeXt V1 (Liu et al., 2022) is used in many works and shows a strong temporal modeling capability in speech domain tasks (Siuzdak, 2023; Okamoto et al., 2024).

As described in Sec.3.1, the model input is character sequence, noisy speech, and masked speech. Before concatenation in the feature dimension, the character sequence first goes through ConvNeXt blocks. Experiments have shown that this way of providing individual modeling space allows text input to better prepare itself before later in-context learning. Unlike the phoneme-level force alignment done in Voicebox, a rigid boundary for text is not explicitly introduced. The semantic and acoustic features are jointly learned with the entire model. Not directly feeding the model with inputs of significant length gap as E2 TTS does, the proposed text refinement mitigates the impact of using inputs with mismatched effective information lengths, despite equal physical length in magnitude as E2 TTS.

The flow step $t$ for CFM is provided as the condition of adaLN-zero rather than appended to the concatenated input sequence in Voicebox. We found that an additional mean pooled token of text sequence for adaLN condition is not essential for the TTS task, maybe because the TTS task requires more rigorously guided results and the mean pooled text token is more coarse.

We adopt some position embedding settings in Voicebox. The flow step is embedded with a sinusoidal position. The concatenated input sequence is added with a convolutional position embedding. We apply a rotary position embedding (RoPE) (Su et al., 2024) for self-attention rather than symmetric bi-directional ALiBi bias (Press et al., 2021). And for extended character sequence $\hat{y}$, we also add it with an absolute sinusoidal position embedding before feeding it into ConvNeXt blocks.

Compared with Voicebox and E2 TTS, we abandoned the U-Net (Ronneberger et al., 2015) style skip connection structure and switched to using DiT with adaLN-zero. Without a phoneme-level duration predictor and explicit alignment process, and nor with extra text encoder and semantically infused neural codec model in DiTTo-TTS, we give the text input a little freedom (individual modeling space) to let it prepare itself before concatenation and in-context learning with speech input.

**Sampling**   As stated in Sec.2.1, the CFM could be viewed as v-prediction with a cosine schedule. For image synthesis, Esser et al. (2024) propose to further schedule the flow step with a single-peak logit-normal (Atchison & Shen, 1980) sampling, in order to give more weight to intermediate flow steps by sampling them more frequently. We speculate that such sampling distributes the model's learning difficulty more evenly over different flow step $t \in [0, 1]$.

In contrast, we train our model with traditional uniformly sampled flow step $t \sim \mathcal{U}[0, 1]$ but apply a non-uniform sampling during inference. In specific, we define a **Sway Sampling** function as

$$f_{sway}(u; s) = u + s \cdot (\cos(\frac{\pi}{2}u) - 1 + u), \tag{7}$$

which is monotonic with coefficient $s \in [-1, \frac{2}{\pi-2}]$. We first sample $u \sim \mathcal{U}[0, 1]$, then apply this function to obtain sway sampled flow step $t$. With $s < 0$, the sampling is sway to left; with $s > 0$, the

sampling is sway to right; and $s = 0$ case equals to uniform sampling. Fig.3 shows the probability density function of Sway Sampling on flow step $t$.

Conceptually, CFM models focus more on sketching the contours of speech in the early stage ($t \rightarrow 0$) from pure noise and later focus more on the embellishment of fine-grained details. Therefore, the alignment between speech and text will be determined based on the first few generated results. With a scale parameter $s < 0$, we make model inference more with smaller $t$, thus providing the ODE solver with more startup information to evaluate more precisely in initial integration steps.

## 4 EXPERIMENTAL SETUP

**Datasets**    We utilize the in-the-wild multilingual speech dataset Emilia (He et al., 2024) to train our base models. After simply filtering out transcription failure and misclassified language speech, we retain approximately 95K hours of English and Chinese data. We also trained small models for ablation study and architecture search on WenetSpeech4TTS (Ma et al., 2024) Premium subset, consisting of a 945 hours Mandarin corpus. Base model configurations are introduced below, and small model configurations are in Appendix B.1. Three test sets are adopted for evaluation, which are LibriSpeech-PC *test-clean* (Meister et al., 2023), Seed-TTS *test-en* (Anastassiou et al., 2024) with 1088 samples from Common Voice (Ardila et al., 2019), and Seed-TTS *test-zh* with 2020 samples from DiDiSpeech (Guo et al., 2021)[2]. Most of the previous English-only models are evaluated on different subsets of LibriSpeech *test-clean* while the used prompt list is not released, which makes fair comparison difficult. Thus we build and release a 4-to-10-second LibriSpeech-PC subset with 1127 samples to facilitate community comparisons.

**Training**    Our base models are trained to 1.2M updates with a batch size of 307,200 audio frames (0.91 hours), for over one week on 8 NVIDIA A100 80G GPUs. The AdamW optimizer (Loshchilov, 2017) is used with a peak learning rate of 7.5e-5, linearly warmed up for 20K updates, and linearly decays over the rest of the training. We set 1 for the max gradient norm clip. The F5-TTS base model has 22 layers, 16 attention heads, 1024/2048 embedding/feed-forward network (FFN) dimension for DiT; and 4 layers, 512/1024 embedding/FFN dimension for ConvNeXt V2; in total 335.8M parameters. The reproduced E2 TTS, a 333.2M flat U-Net equiped Transformer, has 24 layers, 16 attention heads, and 1024/4096 embedding/FFN dimension. Both models use RoPE as mentioned in Sec.3.2, a dropout rate of 0.1 for attention and FFN, the same convolutional position embedding as in Voicebox(Le et al., 2024).

We directly use alphabets and symbols for English, use jieba[3] and pypinyin[4] to process raw Chinese characters to full pinyins. The character embedding vocabulary size is 2546, counting in the special filler token and all other language characters exist in the Emilia dataset as there are many code-switched sentences. For audio samples we use 100-dimensional log mel-filterbank features with 24 kHz sampling rate and hop length 256. A random 70% to 100% of mel frames is masked for infilling task training. For CFG (Sec.2.2) training, first the masked speech input is dropped with a rate of 0.3, then the masked speech again but with text input together is dropped with a rate of 0.2 (Le et al., 2024). We assume that the two-stage control of CFG training may have the model learn more with text alignment.

**Inference**    The inference process is mainly elaborated in Sec.3.1. We use the Exponential Moving Averaged (EMA) (Karras et al., 2024) weights for inference, and the Euler ODE solver for F5-TTS (midpoint for E2 TTS as described in Eskimez et al. (2024)). We use the pretrained vocoder Vocos (Siuzdak, 2023) to convert generated log mel spectrograms to audio signals.

**Baselines**    We compare our models with leading TTS systems including, (mainly) autoregressive models: VALL-E 2 (Chen et al., 2024), MELLE (Meng et al., 2024), FireRedTTS (Guo et al., 2024a) and CosyVoice (Du et al., 2024b); non-autoregressive models: Voicebox (Le et al., 2024), NaturalSpeech 3 (Ju et al., 2024), DiTTo-TTS (Lee et al., 2024), MaskGCT (Wang et al., 2024), Seed-TTS$_{DiT}$ (Anastassiou et al., 2024) and our reproduced E2 TTS (Eskimez et al., 2024). Details of compared models see Appendix A.

---

[2]https://github.com/BytedanceSpeech/seed-tts-eval
[3]https://github.com/fxsjy/jieba
[4]https://github.com/mozillazg/python-pinyin

Table 1: Results on LibriSpeech *test-clean* and LibriSpeech-PC *test-clean*. The boldface indicates the best result, and * denotes the score reported in baseline papers with different subsets for evaluation. The Real-Time Factor (RTF) is computed with the inference time of 10s speech. #Param. stands for the number of learnable parameters and #Data refers to the used training dataset in hours.

| Model | #Param. | #Data(hrs) | WER(%)↓ | SIM-o ↑ | RTF ↓ |
|---|---|---|---|---|---|
| **LibriSpeech *test-clean*** | | | | | |
| Ground Truth (*2.2 hours subset*) | - | - | 2.2* | 0.754* | - |
| VALL-E 2 (Chen et al., 2024) | - | 50K EN | 2.44* | 0.643* | 0.732* |
| MELLE (Meng et al., 2024) | - | 50K EN | 2.10* | 0.625* | 0.549* |
| MELLE-*R2* (Meng et al., 2024) | - | 50K EN | 2.14* | 0.608* | 0.276* |
| Voicebox (Le et al., 2024) | 330M | 60K EN | **1.9*** | **0.662*** | 0.64* |
| DiTTo-TTS (Lee et al., 2024) | 740M | 55K EN | 2.56* | 0.627* | **0.162*** |
| Ground Truth (*40 samples subset*) | - | - | 1.94* | 0.68* | - |
| Voicebox (Le et al., 2024) | 330M | 60K EN | 2.03* | 0.64* | 0.64* |
| NaturalSpeech 3 (Ju et al., 2024) | 500M | 60K EN | **1.94*** | 0.67* | 0.296* |
| MaskGCT (Wang et al., 2024) | 1048M | 100K Multi. | 2.634* | **0.687*** | - |
| **LibriSpeech-PC *test-clean*** | | | | | |
| Ground Truth (*1127 samples 2 hrs*) | - | - | 2.23 | 0.69 | - |
| Vocoder Resynthesized | - | - | 2.32 | 0.66 | - |
| CosyVoice (Du et al., 2024b) | ∼300M | 170K Multi. | 3.59 | 0.66 | 0.92 |
| FireRedTTS (Guo et al., 2024a) | ∼580M | 248K Multi. | 2.69 | 0.47 | 0.84 |
| E2 TTS (32 NFE) (Eskimez et al., 2024) | 333M | 100K Multi. | 2.95 | **0.69** | 0.68 |
| F5-TTS (16 NFE) | 336M | 100K Multi. | 2.53 | 0.66 | **0.15** |
| F5-TTS (32 NFE) | 336M | 100K Multi. | **2.42** | 0.66 | 0.31 |

**Metrics**  We measure the performances under *cross-sentence* task (Wang et al., 2023a; Le et al., 2024). The model is given a reference text, a short speech prompt, and its transcription, and made to synthesize a speech reading the reference text mimicking the speech prompt speaker. In specific, we report Word Error Rate (WER) and speaker Similarity between generated and the original target speeches (SIM-o (Le et al., 2024)) for objective evaluation. For WER, we employ Whisper-large-v3 (Radford et al., 2023) to transcribe English and Paraformer-zh (Gao et al., 2023b) for Chinese, following (Anastassiou et al., 2024). For SIM-o, we use a WavLM-large-based (Chen et al., 2022) speaker verification model to extract speaker embeddings for calculating the cosine similarity of synthesized and ground truth speeches. We use Comparative Mean Opinion Scores (CMOS) and Similarity Mean Opinion Scores (SMOS) for subjective evaluation. For CMOS, human evaluators are given randomly ordered synthesized speech and ground truth, and are to decide how higher the naturalness of the better one surpasses the counterpart, *w.r.t.* prompt speech. For SMOS, human evaluators are to score the similarity between the synthesized and prompt. Details of subjective evaluations can be found in Appendix C.

## 5 EXPERIMENTAL RESULTS

Tab.1 and 2 show the main results of objective and subjective evaluations. We report the average score of three random seed generation results with our model and open-sourced baselines. We use by default a CFG strength of 2 and a Sway Sampling coefficient of −1 for our F5-TTS.

For English zero-shot evaluation, the previous works are hard to compare directly as they use different subsets of LibriSpeech *test-clean* (Panayotov et al., 2015). Although most of them claim to filter out 4-to-10-second utterances as the generation target, the corresponding prompt audios used are not released. Therefore, we build a 4-to-10-second sample test set based on LibriSpeech-PC (Meister et al., 2023) which is an extension of LibriSpeech with additional punctuation marks and casing. To facilitate future comparison, we release the 2-hour test set with 1,127 samples, sourced from 39 speakers (LibriSpeech-PC missing one speaker).

Table 2: Results on two test sets, Seed-TTS *test-en* and *test-zh*. The boldface indicates the best result, the underline denotes the second best, and * denotes scores reported in baseline papers.

| Model | WER(%)↓ | SIM-o ↑ | CMOS ↑ | SMOS ↑ |
|---|---|---|---|---|
| **Seed-TTS *test-en*** | | | | |
| Ground Truth | 2.06 | 0.73 | 0.00 | 3.91 |
| Vocoder Resynthesized | 2.09 | 0.70 | - | - |
| CosyVoice (Du et al., 2024b) | 3.39 | 0.64 | 0.02 | 3.64 |
| FireRedTTS (Guo et al., 2024a) | 3.82 | 0.46 | -1.46 | 2.94 |
| MaskGCT (Wang et al., 2024) | 2.623* | 0.717* | - | - |
| Seed-TTS$_{DiT}$(Anastassiou et al., 2024) | **1.733*** | **0.790*** | - | - |
| E2 TTS (32 NFE) (Eskimez et al., 2024) | 2.19 | 0.71 | 0.06 | 3.81 |
| F5-TTS (16 NFE) | 1.89 | 0.67 | 0.16 | 3.79 |
| F5-TTS (32 NFE) | 1.83 | 0.67 | **0.31** | **3.89** |
| **Seed-TTS *test-zh*** | | | | |
| Ground Truth | 1.26 | 0.76 | 0.00 | 3.72 |
| Vocoder Resynthesized | 1.27 | 0.72 | - | - |
| CosyVoice (Du et al., 2024b) | 3.10 | 0.75 | -0.06 | 3.54 |
| FireRedTTS (Guo et al., 2024a) | 1.51 | 0.63 | -0.49 | 3.28 |
| MaskGCT (Wang et al., 2024) | 2.273* | 0.774* | - | - |
| Seed-TTS$_{DiT}$(Anastassiou et al., 2024) | **1.178*** | **0.809*** | - | - |
| E2 TTS (32 NFE) (Eskimez et al., 2024) | 1.97 | 0.73 | -0.04 | 3.44 |
| F5-TTS (16 NFE) | 1.74 | 0.75 | 0.02 | 3.72 |
| F5-TTS (32 NFE) | 1.56 | 0.76 | **0.21** | **3.83** |

F5-TTS achieves a WER of 2.42 on LibriSpeech-PC *test-clean* with 32 NFE and Sway Sampling, demonstrating its robustness in zero-shot generation. Inference with 16 NFE, F5-TTS gains an RTF of 0.15 while still supporting high-quality generation with a WER of 2.53. It is clear that the Sway Sampling strategy greatly improves performance. The reproduced E2 TTS shows an excellent speaker similarity (SIM) but much worse WER in the zero-shot scenario, indicating the inherent deficiency of alignment robustness.

From the evaluation results on the Seed-TTS test sets, F5-TTS behaves similarly with a close WER to ground truth and comparable SIM scores. It produces smooth and fluent speech in zero-shot generation with a CMOS of 0.31 (0.21) and SMOS of 3.89 (3.83) on Seed-TTS *test-en* (*test-zh*), and surpasses some baseline models trained with larger scales. It is worth mentioning that Seed-TTS with the best result is trained with orders of larger model size and dataset (several million hours) than ours. As stated in Sec.3.1, we simply estimate duration based on the ratio of the audio prompt's transcript length and the text prompt length. If providing ground truth duration, F5-TTS with 32 NFE and Sway Sampling will have WER of 1.74 for *test-en* and 1.53 for *test-zh* while maintaining the same SIM, indicating a high upper bound.

A robustness test on ELLA-V (Song et al., 2024) hard sentences is further included in Appendix B.5. The ablation of vocoders and additional evaluation with a non-PC test set are in Appendix B.6. An analysis of training stability with varying data scales is in Appendix B.7.

## 5.1 ABLATION OF MODEL ARCHITECTURE

To clarify our F5-TTS's efficiency and stress the limitation of E2 TTS. We conduct in-depth ablation studies. We trained small models to 800K updates (each on 8 NVIDIA RTX 3090 GPUs for one week), all scaled to around 155M parameters, on the WenetSpeech4TTS Premium 945 hours Mandarin dataset with half the batch size and the same optimizer and scheduler as base models. Details of small model configurations see Appendix B.1.

We first experiment with pure adaLN DiT (F5-TTS−*Conv2Text*), which fails to learn alignment given simply padded character sequences. Based on the concept of refining the input text represen-

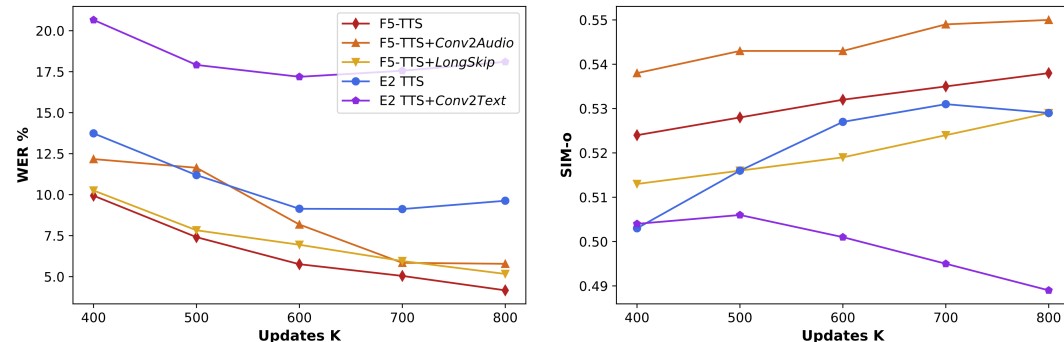

Figure 2: Ablation studies on model architecture. Seed-TTS *test-zh* evaluation results of 155M small models trained with WenetSpeech4TTS Premium a 945 hours Mandarin Corpus.

tation to better align with speech modality, and keep the simplicity of system design, we propose to add jointly learned structure to the input context. Specifically, we leverage ConvNeXt's capabilities of capturing local connections, multi-scale features, and spatial invariance for the input text, which is our F5-TTS. And we ablate with adding the same branch for input speech, denoted F5-TTS+*Conv2Audio*. We further conduct experiments to figure out whether the long skip connection and the pre-refinement of input text are beneficial to the counterpart backbone, *i.e.* F5-TTS and E2 TTS, named F5-TTS+*LongSkip* and E2 TTS+*Conv2Text* respectively. We also tried with the Multi-Modal DiT (MMDiT) (Esser et al., 2024) a double-stream joint-attention structure for the TTS task which learned fast and collapsed fast, resulting in severe repeated utterance with wild timbre and prosody. We assume that the pure MMDiT structure is far too flexible for rigorous task *e.g.* TTS which needs more faithful generation following the prompt guidance.

Fig.2 shows the overall trend of small models' WER and SIM scores evaluated on Seed-TTS *test-zh*. Trained with only 945 hours of data, F5-TTS (32 NFE *w/o* SS) achieves a WER of 4.17 and a SIM of 0.54 at 800K updates, while E2 TTS is 9.63 and 0.53. F5-TTS+*Conv2Audio* trades much alignment robustness (+1.61 WER) with a slightly higher speaker similarity (+0.01 SIM), which is not ideal for scaling up. We found that the long skip connection structure can not simply fit into DiT to improve speaker similarity, while the ConvNeXt for input text refinement can not directly apply to the flat U-Net Transformer to improve WER as well, both showing significant degradation of performance. To further analyze the unsatisfactory results with E2 TTS, we studied the consistent failure (unable to solve with re-ranking) on a 7% of the test set (WER>50%) all along the training process. We found that E2 TTS typically struggles with around 140 samples which we speculate to have a large distribution gap with the train set, while F5-TTS easily tackles this issue.

We investigate the models' behaviors with different input conditions to illustrate the advantages of F5-TTS further and disclose the possible reasons for E2 TTS's deficiency. See from Tab.4 in Appendix B.2, providing the ground truth duration allows more gains on WER for F5-TTS than E2 TTS. By dropping the audio prompt, and synthesizing speech solely with the text prompt, E2 TTS is free of failures. This phenomenon implied a deep entanglement of semantic and acoustic features within E2 TTS's model design. From Tab.3 GFLOPs statistics, F5-TTS carries out faster training and inference than E2 TTS.

The aforementioned limitations of E2 TTS greatly hinder real-world application as the failed generation cannot be solved with re-ranking. Supervised fine-tuning facing out-of-domain data or a tremendous pretraining scale is mandatory for E2 TTS, which is inconvenient for industrial deployment. On the contrary, our F5-TTS better handles zero-shot generation, showing stronger robustness.

## 5.2 ABLATION OF SWAY SAMPLING

It is clear from Fig.3 that a Sway Sampling with a negative $s$ improves the generation results. Further with a more negative $s$, models achieve lower WER and higher SIM scores. We additionally include comparing results on base models with and without Sway Sampling in Appendix B.4.

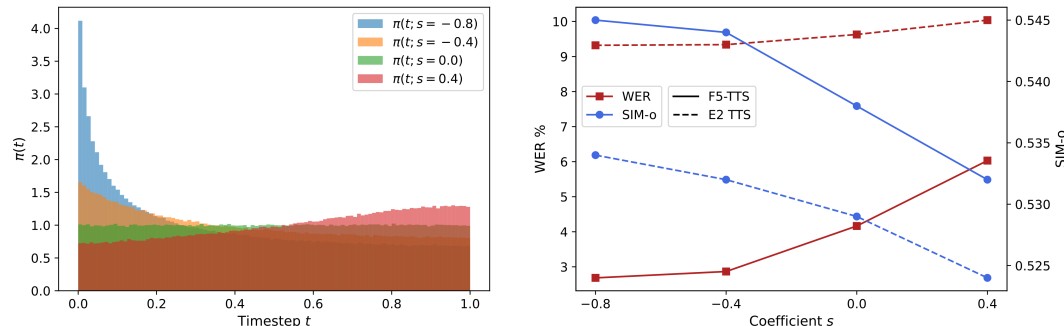

Figure 3: Probability density function of Sway Sampling on flow step $t$ with different coefficient $s$ (left), and small models' performance on Seed-TTS *test-zh* with Sway Sampling (right).

As stated at the end of Sec.3.2, Sway Sampling with $s < 0$ scales more flow step toward early-stage inference ($t \to 0$), thus having CFM models capture more startup information to sketch the contours of target speech better. To be more concrete, we conduct a "leak and override" experiment. We first replace the Gaussian noise input $x_0$ at inference time with a ground-truth-information-leaked input $(1 - t')x_0 + t'x'_{ref}$, where $t' = 0.1$ and $x'_{ref}$ is a duplicate of the audio prompt mel features. Then, we provide a text prompt different from the duplicated audio transcript and let the model continue the subsequent inference (skip the flow steps before $t'$). The model succeeds in overriding leaked utterances and producing speech following the text prompt if Sway Sampling is used, and fails without. Uniformly sampled flow steps will have the model producing speech dominated by leaked information, speaking the duplicated audio prompt's context. Similarly, a leaked timbre can be overridden with another speaker's utterance as an audio prompt, leveraging Sway Sampling.

The experiment result is a shred of strong evidence proving that the early flow steps are crucial for sketching the silhouette of target speech based on given prompts faithfully, the later steps focus more on formed intermediate noisy output, where our sway-to-left sampling ($s < 0$) finds the profitable niche and takes advantage of it. We emphasize that our inference-time Sway Sampling can be easily applied to existing CFM-based models without retraining. And we will work in the future to combine it with training-time noise schedulers and distillation techniques to further boost efficiency.

# 6 CONCLUSION

This work introduces F5-TTS, a fully non-autoregressive text-to-speech system based on flow matching with diffusion transformer (DiT). With a tidy pipeline, literally text in and speech out, F5-TTS achieves state-of-the-art zero-shot ability compared to existing works trained on industry-scale data. We adopt ConvNeXt for text modeling and propose the test-time Sway Sampling strategy to further improve the robustness of speech generation and inference efficiency. Our design allows faster training and inference, by achieving a test-time RTF of 0.15, which is competitive with other heavily optimized TTS models of similar performance. We will open-source our code, and models, to enhance transparency and facilitate reproducible research in this area.

## ETHICS STATEMENTS

This work is purely a research project. F5-TTS is trained on large-scale public multilingual speech data and could synthesize speech of high naturalness and speaker similarity. Given the potential risks in the misuse of the model, such as spoofing voice identification, it should be imperative to implement watermarks and develop a detection model to identify audio outputs.

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

## A  BASELINE DETAILS

**VALL-E 2** (Chen et al., 2024)   A large-scale TTS model shares the same architecture as VALL-E (Wang et al., 2023a) but employs a repetition-aware sampling strategy that promotes more deliberate sampling choices, trained on Libriheavy (Kang et al., 2024) 50K hours English dataset. We compared with results reported in Meng et al. (2024).

**MELLE** (Meng et al., 2024)   An autoregressive large-scale model leverages continuous-valued tokens with variational inference for text-to-speech synthesis. Its variants allow to prediction of multiple mel-spectrogram frames at each time step, noted by MELLE-*Rx* with *x* denotes reduction factor. The model is trained on Libriheavy (Kang et al., 2024) 50K hours English dataset. We compared with results reported in Meng et al. (2024).

**Voicebox** (Le et al., 2024)   A non-autoregressive large-scale model based on flow matching trained with infilling task. We compared with the 330M parameters trained on 60K hours dataset English-only model's results reported in Le et al. (2024) and Ju et al. (2024).

**NaturalSpeech 3** (Ju et al., 2024)   A non-autoregressive large-scale TTS system leverages a factorized neural codec to decouple speech representations and a factorized diffusion model to generate speech based on disentangled attributes. The 500M base model is trained on Librilight (Kahn et al., 2020) a 60K hours English dataset. We compared with scores reported in Ju et al. (2024).

**DiTTo-TTS** (Lee et al., 2024)   A large-scale non-autoregressive TTS model uses a cross-attention Diffusion Transformer and leverages a pretrained language model to enhance the alignment. We compare with DiTTo-en-XL, a 740M model trained on 55K hours English-only dataset, using scores reported in Lee et al. (2024).

**FireRedTTS** (Guo et al., 2024a)   A foundation TTS framework for industry-level generative speech applications. The autoregressive text-to-semantic token model has 400M parameters and the token-to-waveform generation model has about half the parameters. The system is trained with 248K hours of labeled speech data. We use the official code and pre-trained checkpoint to evaluate[5].

**MaskGCT** (Wang et al., 2024)   A large-scale non-autoregressive TTS model without precise alignment information between text and speech following the mask-and-predict learning paradigm. The model is multi-stage, with a 695M text-to-semantic model (T2S) and then a 353M semantic-to-acoustic (S2A) model. The model is trained on Emilia (He et al., 2024) dataset with around 100K Chinese and English in-the-wild speech data. We compare with results reported in Wang et al. (2024).

**Seed-TTS** (Anastassiou et al., 2024)   A family of high-quality versatile speech generation models trained on unknown tremendously large data that is of orders of magnitudes larger than the previously largest TTS systems (Anastassiou et al., 2024). Seed-TTS$_{DiT}$ is a large-scale fully non-autoregressive model. We compare with results reported in Anastassiou et al. (2024).

**E2 TTS** (Eskimez et al., 2024)   A fully non-autoregressive TTS system proposes to model without the phoneme-level alignment in Voicebox, originally trained on Libriheavy (Kang et al., 2024) 50K English dataset. We compare with our reproduced 333M multilingual E2 TTS trained on Emilia (He et al., 2024) dataset with around 100K Chinese and English in-the-wild speech data.

**CosyVoice** (Du et al., 2024b)   A two-stage large-scale TTS system, first autoregressive text-to-token, then a flow matching diffusion model. The model is of around 300M parameters, trained on 170K hours of multilingual speech data. We obtain the evaluation result with the official code and pre-trained checkpoint[6].

---

[5]`https://github.com/FireRedTeam/FireRedTTS`
[6]`https://huggingface.co/model-scope/CosyVoice-300M`

# B    EXPERIMENTAL RESULT SUPPLEMENTS

The UTMOS (Saeki et al., 2022) scores reported in this section are evaluated with an open-source MOS prediction model[7]. The UTMOS is an objective metric measuring naturalness.

## B.1    SMALL MODEL CONFIGURATION

The detailed configuration of small models is shown in Tab.3. In the Transformer column, the numbers denote the Model Dimension, the Number of Layers, the Number of Heads, and the multiples of Hidden Size. In the ConvNeXt column, the numbers denote the Model Dimension, the Number of Layers, and the multiples of Hidden Size. GFLOPs are evaluated using the `thop` Python package.

As mentioned in Sec.3.2, F5-TTS leverages an adaLN DiT backbone, while E2 TTS is a flat U-Net equipped Transformer. F5-TTS+*LongSkip* adds an additional long skip structure connecting the first to the last layer (Lee et al., 2024) in the Transformer. For the Multi-Model Diffusion Transformer (MMDiT) (Esser et al., 2024), a double stream transformer, the setting denotes one stream configuration.

Table 3: Details of small model configurations.

| Model | Transformer | ConvNeXt | #Param. | GFLOPs |
|---|---|---|---|---|
| F5-TTS | 768,18,12,2 | 512,4,2 | 158M | 173 |
| F5-TTS−*Conv2Text* | 768,18,12,2 | - | 153M | 164 |
| F5-TTS+*Conv2Audio* | 768,16,12,2 | 512,4,2 | 163M | 181 |
| F5-TTS+*LongSkip* | 768,18,12,2 | 512,4,2 | 159M | 175 |
| E2 TTS | 768,20,12,4 | - | 157M | 293 |
| E2 TTS+*Conv2Text* | 768,20,12,4 | 512,4,2 | 161M | 301 |
| MMDiT (Esser et al., 2024) | 512,16,16,2 | - | 151M | 104 |

## B.2    ABLATION STUDY ON INPUT CONDITION

The ablation study on different input conditions is conducted with three settings: common input with text and audio prompts, providing ground truth duration information rather than an estimate, and retaining only text input dropping audio prompt. In Tab.4, all evaluations take the 155M small models' checkpoints trained on WenetSpeech4TTS Premium at 800K updates.

Table 4: Ablation study on different input conditions. The boldface indicates the best result, and the underline denotes the second best. All scores are the average of three random seed results.

| Model | Common Input | | Ground Truth Dur. | | Drop Audio Prompt | |
|---|---|---|---|---|---|---|
| | WER ↓ | SIM ↑ | WER ↓ | SIM ↑ | WER ↓ | SIM ↑ |
| F5-TTS | **4.17** | 0.54 | **3.87** | 0.54 | 3.22 | 0.21 |
| F5-TTS+*Conv2Audio* | 5.78 | **0.55** | 5.28 | **0.55** | 3.78 | 0.21 |
| F5-TTS+*LongSkip* | 5.17 | 0.53 | 5.03 | 0.53 | 3.35 | 0.21 |
| E2 TTS | 9.63 | 0.53 | 9.48 | 0.53 | 3.48 | 0.21 |
| E2 TTS+*Conv2Text* | 18.10 | 0.49 | 17.94 | 0.49 | **3.06** | 0.21 |

## B.3    COMPARISON OF ODE SOLVERS

The comparison results of using the Euler (first-order), midpoint (second-order), or improved Heun (third-order, Heun-3) ODE solver during F5-TTS inference are shown in Tab.5. The Euler is inherently faster and performs slightly better typically for larger NFE inference with Sway Sampling (otherwise the Euler solver results in degradation).

---

[7]https://github.com/tarepan/SpeechMOS

Table 5: Evaluation results of F5-TTS on LibriSpeech-PC *test-clean*, Seed-TTS *test-en* and Seed-TTS *test-zh*, employing the Euler, midpoint or Heun-3 ODE solver, and with different Sway Sampling $s$ values. The Real-Time Factor (RTF) is computed with the inference time of 10s speech.

| F5-TTS | LibriSpeech-PC *test-clean* | | | Seed-TTS *test-en* | | | Seed-TTS *test-zh* | | | RTF |
|---|---|---|---|---|---|---|---|---|---|---|
| | WER | SIM-o | UTMOS | WER | SIM-o | UTMOS | WER | SIM-o | UTMOS | |
| Ground Truth | 2.23 | 0.69 | 4.09 | 2.06 | 0.73 | 3.53 | 1.26 | 0.76 | 2.78 | - |
| $s = -1$ | | | | | | | | | | |
| 16 NFE Euler | 2.53 | 0.66 | 3.88 | 1.89 | 0.67 | 3.76 | 1.74 | 0.75 | 2.96 | 0.15 |
| 16 NFE midpoint | 2.43 | 0.66 | 3.87 | 1.88 | 0.66 | 3.70 | 1.61 | 0.75 | 2.87 | 0.26 |
| 32 NFE Euler | 2.42 | 0.66 | 3.90 | 1.83 | 0.67 | 3.76 | 1.56 | 0.76 | 2.95 | 0.31 |
| 32 NFE midpoint | 2.41 | 0.66 | 3.89 | 1.87 | 0.66 | 3.72 | 1.58 | 0.75 | 2.91 | 0.53 |
| 16 NFE Heun-3 | 2.39 | 0.65 | 3.87 | 1.80 | 0.66 | 3.70 | 1.55 | 0.75 | 2.88 | 0.44 |
| $s = -0.8$ | | | | | | | | | | |
| 16 NFE Euler | 2.82 | 0.65 | 3.73 | 2.14 | 0.65 | 3.70 | 2.28 | 0.72 | 2.74 | 0.15 |
| 16 NFE midpoint | 2.58 | 0.65 | 3.86 | 1.86 | 0.65 | 3.68 | 1.70 | 0.73 | 2.83 | 0.26 |
| 32 NFE Euler | 2.50 | 0.66 | 3.89 | 1.81 | 0.67 | 3.74 | 1.62 | 0.75 | 2.94 | 0.31 |
| 32 NFE midpoint | 2.42 | 0.66 | 3.89 | 1.84 | 0.66 | 3.70 | 1.62 | 0.75 | 2.91 | 0.53 |
| 16 NFE Heun-3 | 2.40 | 0.65 | 3.85 | 1.78 | 0.66 | 3.68 | 1.56 | 0.74 | 2.84 | 0.44 |

Table 6: Base model evaluation results on LibriSpeech-PC *test-clean*, Seed-TTS *test-en* and *test-zh*, with and without proposed test-time Sway Sampling (SS, with coefficient $s = -1$) strategy for flow steps. All generations leverage the midpoint ODE solver for ease of ablation.

| Model | WER(%)↓ | SIM-o ↑ | UTMOS ↑ | RTF ↓ |
|---|---|---|---|---|
| **LibriSpeech-PC *test-clean*** | | | | |
| Ground Truth (*1127 samples*) | 2.23 | 0.69 | 4.09 | - |
| Vocoder Resynthesized | 2.32 | 0.66 | 3.64 | - |
| E2 TTS (16 NFE *w/* SS) | 2.86 | 0.71 | 3.66 | 0.34 |
| E2 TTS (32 NFE *w/* SS) | 2.84 | 0.72 | 3.70 | 0.68 |
| E2 TTS (32 NFE *w/o* SS) | 2.95 | 0.69 | 3.56 | 0.68 |
| F5-TTS (16 NFE *w/* SS) | 2.43 | 0.66 | 3.87 | 0.26 |
| F5-TTS (32 NFE *w/* SS) | 2.41 | 0.66 | 3.89 | 0.53 |
| F5-TTS (32 NFE *w/o* SS) | 2.84 | 0.62 | 3.70 | 0.53 |
| **Seed-TTS *test-en*** | | | | |
| Ground Truth (*1088 samples*) | 2.06 | 0.73 | 3.53 | - |
| Vocoder Resynthesized | 2.09 | 0.70 | 3.33 | - |
| E2 TTS (16 NFE *w/* SS) | 1.99 | 0.72 | 3.55 | 0.34 |
| E2 TTS (32 NFE *w/* SS) | 1.98 | 0.73 | 3.57 | 0.68 |
| E2 TTS (32 NFE *w/o* SS) | 2.19 | 0.71 | 3.33 | 0.68 |
| F5-TTS (16 NFE *w/* SS) | 1.88 | 0.66 | 3.70 | 0.26 |
| F5-TTS (32 NFE *w/* SS) | 1.87 | 0.66 | 3.72 | 0.53 |
| F5-TTS (32 NFE *w/o* SS) | 1.93 | 0.63 | 3.51 | 0.53 |
| **Seed-TTS *test-zh*** | | | | |
| Ground Truth (*2020 samples*) | 1.26 | 0.76 | 2.78 | - |
| Vocoder Resynthesized | 1.27 | 0.72 | 2.61 | - |
| E2 TTS (16 NFE *w/* SS) | 1.80 | 0.78 | 2.84 | 0.34 |
| E2 TTS (32 NFE *w/* SS) | 1.77 | 0.78 | 2.87 | 0.68 |
| E2 TTS (32 NFE *w/o* SS) | 1.97 | 0.73 | 2.49 | 0.68 |
| F5-TTS (16 NFE *w/* SS) | 1.61 | 0.75 | 2.87 | 0.26 |
| F5-TTS (32 NFE *w/* SS) | 1.58 | 0.75 | 2.91 | 0.53 |
| F5-TTS (32 NFE *w/o* SS) | 1.93 | 0.69 | 2.58 | 0.53 |

### B.4 SWAY SAMPLING EFFECTIVENESS ON BASE MODELS

From Tab.6, it is clear that our Sway Sampling strategy for test-time flow steps consistently improves the zero-shot generation performance in aspects of faithfulness to prompt text (WER), speaker similarity (SIM), and naturalness (UTMOS). The gain of applying Sway Sampling to E2 TTS (Eskimez et al., 2024) proves that our Sway Sampling strategy is universally applicable to existing flow matching based TTS models.

### B.5 ELLA-V HARD SENTENCES EVALUATION

ELLA-V (Song et al., 2024) proposed a challenging set containing 100 difficult textual patterns evaluating the robustness of the TTS model. Following previous works (Chen et al., 2024; Meng et al., 2024; Eskimez et al., 2024), we include generated samples in our demo page[8]. We additionally compare our model with the objective evaluation results reported in E1 TTS (Liu et al., 2024b).

StyleTTS 2 is a TTS model leveraging style diffusion and adversarial training with large speech language models. CosyVoice is a two-stage large-scale TTS system, consisting of a text-to-token AR model and a token-to-speech flow matching model. Concurrent with our work, E1 TTS$_{DMD}$ is a diffusion-based NAR model with a distribution matching distillation technique to achieve one-step TTS generation. Since the prompts used by E1 TTS$_{DMD}$ are not released, we randomly sample 3-second-long speeches in our LibriSpeech-PC *test-clean* set as audio prompts. The evaluation result is in Tab.7. We evaluate the reproduced E2 TTS and our F5-TTS with 32 NFE and Sway Sampling and report the averaged score of three random seed results.

Table 7: Results of zero-shot TTS WER on ELLA-V hard sentences. The asterisk * denotes the score reported in E1 TTS. Sub. for substitution, Del. for Deletion, and Ins. for Insertion.

| Model | WER(%))↓ | Sub.(%)↓ | Del.(%)↓ | Ins.(%)↓ |
|---|---|---|---|---|
| StyleTTS 2 (Li et al., 2024b) | 4.83* | 2.17* | 2.03* | 0.61* |
| CosyVoice (Du et al., 2024b) | 8.30* | 3.47* | 2.74* | 1.93* |
| E1 TTS$_{DMD}$ (Liu et al., 2024b) | 4.29* | 1.89* | 1.62* | 0.74* |
| E2 TTS (Eskimez et al., 2024) | 8.58 | 3.70 | 4.82 | 0.06 |
| F5-TTS | 4.40 | 1.81 | 2.40 | 0.18 |

We note that a higher WER compared to the results on commonly used test sets is partially due to mispronunciation (*yogis* to *yojus*, *cavorts* to *caverts*, *etc.*). The high Deletion rate indicates a word-skipping phenomenon when our model encounters a stack of repeating words. The low Insertion rate makes it clear that our model is free of endless repetition. We further emphasize that prompts from different speakers will spell very distinct utterances, where the ASR model transcribes correctly for one, and fails for another (*e.g. quokkas* to *Cocos*).

### B.6 COMPARISON OF VOCODERS AND BETWEEN PC AND NON-PC

The inference results with pretrained BigVGAN (Lee et al., 2022) and Vocos (Siuzdak, 2023) respectively as vocoder are shown in Tab.8, along with additional evaluation on a non-Capitalized version removing all Punctuations (non-PC) of the filtered LibriSpeech-PC (LS-PC) *test-clean* subset. The non-PC version equals a LibriSpeech (LS) *test-clean* subset, with which we provide more comprehensive comparisons with previous works.

Moreover, we include WER scores measuring with a Hubert-large-based (Hsu et al., 2021) ASR model[9], with which our reproduced multilingual E2 TTS with 32 NFE and Vocos as vocoder achieves a WER of 2.92 on LS-PC *test-clean* and 2.66 if Sway Sampling applied.

---

[8]https://F5-TTS.github.io
[9]https://huggingface.co/facebook/hubert-large-ls960-ft

Table 8: F5-TTS Base model evaluation results on LS-PC *test-clean*, LS *test-clean*, Seed-TTS *test-en* (Seed *test-en*) and *test-zh* (Seed *test-zh*) with BigVGAN and Vocos, default setting as in Sec.5. The WER scores in brackets indicate results leveraging the Hubert-large-based ASR model.

| NFE steps | LS-PC *test-clean* | | LS *test-clean* | | Seed *test-en* | | Seed *test-zh* | |
| & Vocoder | WER | SIM-o | WER | SIM-o | WER | SIM-o | WER | SIM-o |
|---|---|---|---|---|---|---|---|---|
| Ground Truth | 2.23(1.89) | 0.69 | 2.29(1.86) | 0.69 | 2.06 | 0.73 | 1.26 | 0.76 |
| 16 NFE - Vocos | 2.53(2.34) | 0.66 | 2.72(2.53) | 0.66 | 1.89 | 0.67 | 1.74 | 0.75 |
| 16 NFE - BigVGAN | 2.21(1.96) | 0.67 | 2.55(2.34) | 0.67 | 1.65 | 0.66 | 1.64 | 0.74 |
| 32 NFE - Vocos | 2.42(2.09) | 0.66 | 2.44(2.16) | 0.66 | 1.83 | 0.67 | 1.56 | 0.76 |
| 32 NFE - BigVGAN | 2.11(1.81) | 0.67 | 2.28(2.03) | 0.67 | 1.62 | 0.66 | 1.53 | 0.74 |

## B.7 TRAINING STABILITY WITH DIFFERENT DATASET SCALES

We train F5-TTS 158M small models on LibriTTS (Zen et al., 2019) 585 hours and LJSpeech (Ito & Johnson, 2017) 24 hours English datasets to provide insights on our model's training stability with different dataset scales, typically to see whether it can maintain stable training on limited data. Both training takes place with the same configuration as described in Sec.5.1 and Appendix B.1 despite a batch size of 307,200 audio frames (0.91 hours) as base models. Every 100K update takes approximately 8 hours on 8 NVIDIA H100 SXM GPUs.

Same as Sec.5, we report the average score of three random seed generation results, using a CFG strength of 2, a Sway Sampling coefficient of $-1$, and 32 NFE steps. Since LJSpeech is a single-speaker dataset, we measure the metrics on in-set tests (1000 samples organized with 4 to 7 seconds to infer with an around 3-second prompt). It is clear from Tab.9 that our design enables stable training to learn speech-text alignment (without grapheme-to-phoneme) with varying data amounts.

Table 9: F5-TTS small models evaluation results on LibriSpeech-PC *test-clean* (model trained on LibriTTS 585 hours multi-speaker dataset), and on LJSpeech in-set test samples (model trained on 24 hours single-speaker LJSpeech); Vocos as vocoder, Whisper-large-v3 as ASR model.

| Train Set | LibriTTS - 585 hours | | | LJSpeech - 24 hours | | |
| Test Set | LibriSpeech-PC *test-clean* | | | LJSpeech in-set tests | | |
| Updates | WER(%)↓ | SIM-o ↑ | UTMOS ↑ | WER(%)↓ | SIM-o ↑ | UTMOS ↑ |
|---|---|---|---|---|---|---|
| Ground Truth | 2.23 | 0.69 | 4.09 | 2.36 | 0.72 | 4.36 |
| 100K | 29.5 | 0.53 | 3.78 | 5.64 | 0.72 | 4.17 |
| 200K | 4.58 | 0.59 | 4.07 | 2.93 | 0.72 | 4.18 |
| 300K | 2.71 | 0.60 | 4.11 | 3.26 | 0.71 | 4.12 |
| 400K | 2.44 | 0.60 | 4.11 | 3.90 | 0.70 | 4.05 |
| 500K | 2.20 | 0.60 | 4.10 | 4.68 | 0.70 | 3.99 |
| 600K | 2.23 | 0.59 | 4.10 | 5.25 | 0.69 | 3.93 |

## C  SUBJECTIVE EVALUATION DETAILS

To evaluate speech quality, we conduct a CMOS subjective evaluation. 20 natives were invited for both English and Mandarin to evaluate 30 rounds with randomly selected utterances for all three test sets and all model variants. Evaluators were informed in detail about the guidelines and scoring criteria for the CMOS test, for example, the general instruction part:

- Most important: use high-quality studio headphones and a good sound card!
- Listen through all test files and test sets before you do any ratings to get used to the material.
- Rate the quality of the test items only compared to the reference on top.
- Try to rate the overall impression of a test item and don't concentrate on single aspects.

For the CMOS test, the overall quality of a generated speech is first rated from $-3$ (bad quality compared to the reference) to $+3$ (much better than the reference) integer scale, then reported in average differentials with received scores of ground truth speech. For SMOS, a 1 to 5 with 0.5 as an interval rating is employed (higher better). Judges are to score the similarity between the synthesized and prompt speech with clearly differentiated instructions mentioning:

- Try to rate concentrating on the speaker similarity aspects with reference speech.

We encourage more rigorous and transparent subjective evaluations, such as releasing used samples if not open-sourcing the model checkpoints. Meanwhile inviting more evaluators leads to more comprehensive and fair rating scores. Just for reference, DiTTo-TTS (Lee et al., 2024) received and reported 6 and 12 ratings for SMOS and CMOS, respectively, NaturalSpeech 3 (Ju et al., 2024) invited 12 natives to judge 20 samples for CMOS and 10 samples for SMOS.

