# OpenReview forum: "F5-TTS: A Fairytaler that Fakes Fluent and Faithful Speech with Flow Matching"
_ICLR.cc/2025/Conference — ICLR 2025 Conference Withdrawn Submission_

### Official Review · Reviewer_X8VX · 2024-10-27

**Soundness:** 3
**Presentation:** 3
**Contribution:** 3
**Rating:** 8
**Confidence:** 3

**Summary:**

This study uses flow-matching based diffusion model for TTS. As opposed to the earlier studies which usually require additional modules such as an alignment model (e.g. in Voicebox), the proposed model pads the character sequence up to the length of the speech features and uses a ConvNext-like model to let the text features align with speech during the training process. The paper also proposes to use Sway sampling during model inference. This strategy samples the timestep non-uniformly. According to the experiments on Librispeech and a Chinese set, the proposed model results in comparable WER on the test data with previous models (Voicebox and E2-TTS). The RTF also improves in the proposed model.

After reading the rebuttal, I would like to maintain my score.

**Strengths:**

Originality,

- Sufficiently original: The paper builds on the idea of E2-TTS which pads the character sequence up to the number of speech feature frames and with different modeling choices (transformer instead of Unet, and ConveNext based text frontend), the proposed system can handle better speech and text alignment.

- The paper also proposes Sway sampling strategy to the inference instead of uniform time step sampling. The paper suggests that one advantage of this strategy is that it can prevent speaker information leaks in the cases where a speaker or context change occurs.

- The way the paper uses time step representation in the training setup is different than prior studies.

Quality,

- Several ablation experiments are performed. Some of those results are included in the Appendix.


Clarity,

- Writing is clear.

- Experimental setup details are mentioned in detail.


Significance

- Not requiring additional modules (e.g. a duration model) is an advantage of the proposed method.

**Weaknesses:**

1. One problem in experimental settings is the lack of direct comparison between Voicebox, E2-TTS and the current study. The paper already mentions that the prompting test sets are not known and hence it is hard to compare, a reproduction of the setups could have solved this problem. In the Appendix, it seems that there are some comparisons with E2. It would be better to include some of those comparisons in the main text.

2. Table 1, the upper section related to Librispeech test-clean does not show the proposed system's results.

3. There are some minor issues in table captions: Tables 1 and 2 mention that (w/o SS) means without Sway Sampling, but this phrase is not used anywhere in these tables.

**Questions:**

1. Even though the supplemental results show WER and RTF improvements over E2-TTS, in the main text, it appears F5 performs on par with previous methods. Especially, in Table 1, the WER seems to be worse than the closely related two models. Could you please explain the comparison more clearly in the main text?

2. Regarding the speaker similarity metrics vs. the ablation in Section 5.2: Do the authors think that Sway sampling has the advantage which is mentioned in Section 5.2 but at the same time hurting speaker similarity to some extend?

3. Since in Table 1 the paper compares the punctuation added and capitalized version of Librispeech with the widely known original version, have the authors tried a model without including punctuation in their vocabulary? If yes, how does it compare to the presented results?

**Details Of Ethics Concerns:**

Nothing specific to this study except the regular concerns around TTS models as indicated in the paper.

---

> ### Author Response · Authors · 2024-11-20
> **Response to Reviewer X8VX [1/2]**
>
> We sincerely appreciate your positive review and constructive comments. Below we address the concerns mentioned in the review.
>
> > **W1,2**: Lack of direct comparison with previous works and on LibriSpeech test-clean.
>
> Due to space constraints, we put the comparison in the Appendix. Since the setup of prompts in previous works' test sets remains unknown, we take the LibriSpeech-PC (LS-PC) 4~10s filtered set (which is organized in the same way as in the E2 TTS paper, despite having differently paired reference audios for the reason mentioned above) to compare with reproduced E2 TTS and other open-source models.
>
> Meanwhile, we additionally conduct evaluations on a non-capitalized LS-PC 4\~10s test set while removing all punctuations, which can also be viewed as a LibriSpeech test-clean 4~10s subset. A pity is that the LS-PC misses one speaker, though it still serves as a solid set of 1127 samples in total 2 hours, enabling future works to switch from a PC or non-PC version with the same reference audio pairs.
>
> Part of updated evaluation results (Hubert-large-based ASR model to transcribe, results in brackets are with Whisper-large-v3 as before, more results in Appendix B.6), asterisk denotes scores reported in E2 TTS paper (both are English-only models trained on Libriheavy 50K hours):
>
> | Model                          | LS-PC test-clean WER | SIM | LibriSpeech test-clean WER | SIM |
> |--------------------------------|--------------------------------:|-------------------------------|----------------------------:|----------------------------|
> | Ground Truth (E2 TTS subset)   | 2.0*                           | 0.695*                        | /                          | /                          |
> | Voicebox (32 NFE) - BigVGAN     | 2.2*                           | 0.667*                        | /                          | /                          |
> | E2 TTS (32 NFE) - BigVGAN       | 2.0*                           | 0.675*                        | /                          | /                          |
> | Ground Truth (our subset)       | 1.89 (2.23)                    | 0.69                          | 1.86 (2.29)                | 0.69                       |
> | F5-TTS (32 NFE) - Vocos         | 2.09 (2.42)                    | 0.66                          | 2.16 (2.44)                | 0.66                       |
> | F5-TTS (32 NFE) - BigVGAN       | 1.81 (2.11)                    | 0.67                          | 2.03 (2.28)                | 0.67                       |
>
> > **W3**: Redundant *(w/o SS)* in Tables 1 and 2.
>
> Many thanks for pointing out this, we have handled it in the updated version.
>
> > **Q1**: Worse WER compared to closely related two models in Table 1.
>
> In Table 1, we are making efforts toward a thorough comparison with existing outstanding models. To be specific, we divide the table into 3 subsets to avoid any ambiguity as to the fact of different reference audio pairs posed in each subset (otherwise we could take the scores in the above table to compare).
>
> From the relative result compared to each subset's ground truth score, we are confident that our F5-TTS shows no worse performance. Moreover, since we leveraged a multilingual in-the-wild dataset for training which contains not typically high quality and uncensored speech-text pairs compared to the English-only audiobook corpus leveraged by other models, there will be broader space for F5-TTS to improve on English tasks.
>
> > **Q2**: Will Sway sampling have the advantage but at the same time hurting speaker similarity to some extent?
>
> We appreciate this highly observant question. From the overall objective metric, Sway Sampling always improves the performance (more rigorously, Sway Sampling with (more) negative coefficient). We have designed the function so that the timesteps from the larger side will not vanish (still holds several NFE steps there, not deviate too much from the training pattern) even with the most aggressive Sway Sampling case.
>
> While in practice, we observed a slight increase in the probability of some artifacts (e.g. exaggerated fricative), when using a more aggressive Sway Sampling strategy with the same classifier-free-guidance (CFG) coefficient (2.0 in our paper). Intuitively, a more aggressive Sway Sampling simulates a uniform sampling with more NFE on the small timestep side, while the common practice of previous works is to search for the most suitable CFG coefficient accordingly. Due to main text space limitations, we did not perform meticulous ablation of CFG choice but here we are willing to share this observation.

---

> > ### Author Response · Authors · 2024-11-20
> > **Response to Reviewer X8VX [2/2]**
> >
> > > **Q3**: Evaluation on original non-PC Librispeech and comparison with current results on PC version.
> >
> > We include our base model results on the non-PC LibriSpeech test-clean set above and more statistics in Appendix B.6. A slight degradation will occur when our base model is posed with a different format input to the training stage. We believe that training/finetuning with corpus excluding punctuations will address this.
> >
> > We would also like to share some observations that may not just fit in the paper but are interesting, that including the punctuations enables introducing explicit pause with "," ".", direct pronunciation of "\%" as "percent" with no need of normalization, enhancement of interrogative tone to some extent with "?", etc.

---

> > > ### Comment · Reviewer_X8VX · 2024-11-24
> > >
> > > I would like to thank the author(s) for responding to my questions and sharing further insights about the model.
> > >
> > > I have also read the rest of the reviews and I would like to keep my score as is.

---

> > > > ### Author Response · Authors · 2024-11-25
> > > >
> > > > Again, we would like to express our sincere thanks for your positive review and insightful suggestions.
> > > >
> > > > We truly value your time and efforts in the reviewing process.

---

### Official Review · Reviewer_7LV6 · 2024-10-31

**Soundness:** 3
**Presentation:** 3
**Contribution:** 2
**Rating:** 6
**Confidence:** 4

**Summary:**

The authors propose F5-TTS, a zero-shot text-to-speech model that operates without explicit alignment or a pre-trained text encoder. Key distinguishing features from previous research include: (1) the use of a DiT-based structure as an estimator, (2) an additional module (ConvNeXt) for character encoding, and (3) the introduction of the Sway Sampling strategy to improve sampling efficiency. The model demonstrates respectable speaker similarity, audio quality, and pronunciation accuracy in English and Chinese benchmarks. The necessity of the proposed components is further substantiated through ablation studies.

**Strengths:**

- The proposed elements are clearly articulated, and each is thoroughly examined through ablation studies.

- The authors have made commendable efforts to compare their model against a wide range of baselines and benchmarks. Recently, many zero-shot TTS models, particularly those trained on large-scale datasets, have refrained from open-sourcing for various reasons, making it challenging to evaluate new models. In this regard, it is a distinct strength that the authors not only disclose their evaluation data and methods, such as LibriSpeech-PC, but also conduct comprehensive comparisons with diverse models.

- Similarly, the paper states that they plan to release their code and checkpoints soon, which is another clear advantage. Given the limited availability of large-scale models, making their code and model accessible would greatly benefit other researchers in the field.

**Weaknesses:**

- While I do not personally doubt the effectiveness of the proposed methodology, the contributions feel somewhat limited. This is not intended to downplay the model's design and structure for stable training. However, as the authors themselves mention, adopting a DiT structure or performing encoding to compress textual information has been already proposed in previous studies.

- Regarding Sway Sampling, a richer comparison with other sampling strategies could have highlighted its effectiveness more distinctly, especially given the recent influx of sampling techniques based on flow matching and diffusion. In addition to a thorough hyperparameter search within Sway Sampling or evaluating the effects of switching it on and off, comparisons with alternative sampling strategies would have further strengthened the impact of the proposed method.

**Questions:**

- I would like to inquire if comparisons with other sampling strategies from the TTS field or related domains could be provided—starting from more straightforward methods like the Heun sampler and beyond.

- Given the lightweight approach to encoding text using an additional layer (ConvNeXt) compared to a language model, my impression is that this method may require a substantial amount of data to robustly learn pronunciation and speech-text alignment. As the authors highlight the stability of alignment training as a strength of this work, I am curious to know the lower limits of the data threshold while still maintaining stable training (although I understand that the paper's primary focus may not be on stability relative to data quantity; this question is posed out of pure curiosity). For instance, would it be feasible to achieve stable character-speech alignment training with a single-speaker dataset such as LJSpeech, which contains around 20 hours of data? Additionally, how much training time is typically needed to achieve stable pronunciation accuracy? Including results that reflect the impact of varying training data amounts would add valuable insights.

If these questions are addressed, I would like to increase the rating.

**Details Of Ethics Concerns:**

The authors have provided comments on this topic following the Conclusion section.

---

> ### Author Response · Authors · 2024-11-20
> **Response to Reviewer 7LV6 [1/2]**
>
> We sincerely appreciate your careful reading and providing constructive feedback. Below we address the concerns mentioned in the review.
>
> > **W1**: Limited contributions as a DiT structure or text encoder already proposed in previous studies.
>
> We acknowledge that we have not made radical innovations in proposing a new backbone structure, but we believe that our work has the following sufficient contributions in the scope of generative models:
>
> 1. We first prove the feasibility of a medium-scale (336M) adaptive layernorm DiT model with effective ConvNeXt for text refinement on the zero-shot text-to-speech task. To be specific, we achieve comparable performance with the in-context-learning pattern compared to the cross-attention structure nor with explicit phoneme-level alignment, and it is with a much smaller model size compared to recent DiT-based works.
>
> 2. We also arranged a multilingual training setup with a large-scale in-the-wild but uncensored corpus, which was organized with the Whisper medium for language identification and transcription. On the one hand, our work achieves comparable performance with a relatively small model size compared to other models (many are English-only models trained on high-quality audiobook datasets) on English-only tasks. On the other hand, we conduct evaluations on more languages (here Mandarin) and verify our model's seamless code-switching ability (no cherry-picked samples on the demo page) which are also contributions made promoting and exploring capacities under the new paradigm of using sentence-level duration.
>
> 3. The Sway Sampling strategy is proposed to improve sampling efficiency and can be easily applied to other flow matching models, which just fit our simple and effective scheme.
>
> 4. We believe that less is more. We contribute and make transparent a simple and effective design pointing out that a modest refinement for text can allow good results while bringing a tidy pipeline, fast training, and fast inference which are valuable qualities within the scope of generative models.
>
> > **W2,Q1**: A richer comparison between Sway Sampling and other sampling strategies.
>
> Thanks for the constructive suggestion. We have updated Appendix B.4 with comparison results of Sway Sampling incorporating
> - the Euler ODE solver (first-order; default in our updated version, for F5-TTS base model's 1.2M updates result in Table 1 and 2),
> - the midpoint ODE solver (second-order; used by Voicebox, E2 TTS, and our previous version of F5-TTS),
> - the Heun-3 ODE solver (third-order improved Heun; to make a difference with second-order solver),
>
> while switching off Sway Sampling equals to uniformly sampled timestep at inference time used by previous TTS models. From the results, Sway Sampling further allows switching from 2nd-order midpoint to 1st-order Euler which brings significant speed up and maintains the performance.
>
> | F5-TTS, s=-1       | RTF | LS-PC test-clean WER | SIM | Seed-TTS test-en WER | SIM | Seed-TTS test-zh WER | SIM|
> |--------------------|-----------------------:|----------------------:|----------------------|----------------------:|----------------------|----------------------:|-----|
> | Ground Truth           |  -     | 2.23                  | 0.69                 | 2.06                 | 0.73                 | 1.26                 | 0.76                 |
> | 16 NFE - Euler        | 0.15 | 2.53                  | 0.66                 | 1.89                 | 0.67                 | 1.74                 | 0.75                 |
> | 16 NFE - midpoint   | 0.26 | 2.43                  | 0.66                 | 1.88                 | 0.66                 | 1.61                 | 0.75                 |
> | 32 NFE - Euler        | 0.31 | 2.42                  | 0.66                 | 1.83                 | 0.67                 | 1.56                 | 0.76                 |
> | 32 NFE - midpoint   | 0.53 | 2.41                  | 0.66                 | 1.87                 | 0.66                 | 1.58                 | 0.75                 |
> | 16 NFE - Heun3      | 0.44 | 2.39                  | 0.65                 | 1.80                 | 0.66                 | 1.55                 | 0.75                 |

---

> ### Author Response · Authors · 2024-11-20
> **Response to Reviewer 7LV6 [2/2]**
>
> > **Q2**: The lower limits of the data threshold while still maintaining stable training
>
> We are pleased to share our relevant experimental results here and have updated them in our paper.  We train F5-TTS 158M small models on LibriTTS 585 hours and LJSpeech 24-hour English datasets to provide some insights. The model trained on LibriTTS is evaluated on LibriSpeech-PC test-clean. The model trained on LJSpeech is evaluated on in-set 1000 samples since LJSpeech is an up-to-10-second single-speaker dataset.
>
> | F5-TTS small | Trained on LibriTTS WER | SIM | Trained on LJSpeech WER | SIM |
> |--------------|--------------------------:|-------------------------|-------------------------:|-------------------------|
> | Ground Truth | 2.23                     | 0.69                    | 2.36                    | 0.72                    |
> | 100K updates | 29.5                     | 0.53                    | 5.64                    | 0.72                    |
> | 200K updates | 4.58                     | 0.59                    | 2.93                    | 0.72                    |
> | 300K updates | 2.71                     | 0.60                    | 3.26                    | 0.71                    |
> | 400K updates | 2.44                     | 0.60                    | 3.90                    | 0.70                    |
> | 500K updates | 2.20                     | 0.60                    | 4.68                    | 0.70                    |
> | 600K updates | 2.23                     | 0.59                    | 5.25                    | 0.69                    |
>
> `How much training time is typically needed to achieve stable pronunciation accuracy?`
>
> Every 100K update takes around 8 hours on 8 NVIDIA H100 SXM GPUs. Thus 40 hours on LibriTTS to achieve pretty good zero-shot results (an even lower WER than ground truth, and an acceptable SIM with only 585 hours); and 16 hours on the single-speaker LJSpeech to learn character-speech alignment and close voice-cloning.

---

> > ### Comment · Reviewer_7LV6 · 2024-11-21
> > **Thank you for your kind response.**
> >
> > Thank you for your kind response.
> >
> > Personally, I believe that the sampling scheme plays a crucial role not only in SIM and pronunciation accuracy but particularly in audio quality. Considering the cost and time involved, obtaining MOS might be challenging, but I am curious about the results of UT-MOS or other neural network-based mean opinion score predictions. May I kindly ask if it would be possible to include UT-MOS results for the first set of experiments (W2, Q1)?
> >
> > Separately, my questions have been resolved, thanks to your detailed explanation. It seems that the robustness of F5-TTS to dataset size is one of its strengths. As mentioned earlier, I will slightly increase my score.

---

> ### Author Response · Authors · 2024-11-21
> **Response to Reviewer 7LV6 (Cont.)**
>
> We appreciate your feedback and are very willing to include the results of the UTMOS evaluation. We have also updated the paper with proper citations of UTMOS and the used repository.
>
> | Model | RTF| LS-PC test-clean WER | SIM | UTMOS | Seed-TTS test-en WER | SIM | UTMOS | Seed-TTS test-zh WER | SIM | UTMOS |
> |-----|-----:|------:|-----|-------|-----:|-----|-----|-------:|-----|-----|
> | Ground Truth             |  -  | 2.23  | 0.69 |  4.09 | 2.06 | 0.73 | 3.53 | 1.26  | 0.76 | 2.78 |
> | Vocoder Resyn.         |  -  | 2.32  | 0.66 |  3.64 | 2.09 | 0.70 | 3.33 | 1.27  | 0.72 | 2.61 |
> | **F5-TTS $s=-1$ (*w/ SS*)**  | | | | | | | |
> | 16 NFE - Euler       | 0.15 | 2.53 | 0.66 | **3.88** | 1.89 | 0.67 | **3.76** | 1.74 | 0.75 | **2.96** |
> | 16 NFE - midpoint   | 0.26 | 2.43 | 0.66 | 3.87 | 1.88 | 0.66 | 3.70 | 1.61 | 0.75 | 2.87 |
> | 32 NFE - Euler        | 0.31 | 2.42 | 0.66 | 3.90 | 1.83 | 0.67 | 3.76 | 1.56 | 0.76 | 2.95 |
> | 32 NFE - midpoint   | 0.53 | 2.41 | 0.66 | 3.89 | 1.87 | 0.66 | 3.72 | 1.58 | 0.75 | 2.91 |
> | 16 NFE - Heun3      | 0.44 | 2.39 | 0.65 | 3.87 | 1.80 | 0.66 | 3.70 | 1.55 | 0.75 | 2.88 |
> | **F5-TTS $s=-0.8$**  | | | | | | | |
> | 16 NFE - Euler       | 0.15 | 2.82 | 0.65 | **3.73** | 2.14 | 0.65 | **3.70** | 2.28 | 0.72 | **2.74** |
> | 16 NFE - midpoint   | 0.26 | 2.58 | 0.65 | 3.86 | 1.86 | 0.65 | 3.68 | 1.70 | 0.73 | 2.83 |
> | 32 NFE - Euler        | 0.31 | 2.50 | 0.66 | 3.89 | 1.81 | 0.67 | 3.74 | 1.62 | 0.75 | 2.94 |
> | 32 NFE - midpoint   | 0.53 | 2.42 | 0.66 | 3.89 | 1.84 | 0.66 | 3.70 | 1.62 | 0.75 | 2.91 |
> | 16 NFE - Heun3      | 0.44 | 2.40 | 0.65 | 3.85 | 1.78 | 0.66 | 3.68 | 1.56 | 0.74 | 2.84 |
> | **F5-TTS $s=0$ (*w/o SS*)**  | | | | | | | |
> | 32 NFE - midpoint   | 0.53 | 2.84 | 0.62 | 3.70 | 1.93 | 0.63 | 3.51 | 1.93 | 0.69 | 2.58 |
> | **E2 TTS Midpoint**  | | | | | | | |
> | 16 NFE - *w/ SS*      | 0.34 | 2.86 | 0.71 | 3.66 | 1.99 | 0.72 | 3.55 | 1.80 | 0.78 | 2.84 |
> | 32 NFE - *w/ SS*      | 0.68 | 2.84 | 0.72 | 3.70 | 1.98 | 0.73 | 3.57 | 1.77 | 0.78 | 2.87 |
> | 32 NFE - *w/o SS*    | 0.68 | 2.95 | 0.69 | 3.56 | 2.19 | 0.71 | 3.33 | 1.97 | 0.73 | 2.49 |
>
> The UTMOS results confirm with CMOS in the main text, and the differences between switching on and off Sway Sampling (SS) add to the effectiveness. It is quite interesting that the simplest 1st-order Euler ODE and the most aggressive SS supplement each other better (in terms of performance and efficiency; ***bold***).
>
> We also include UTMOS results for small models trained with limited data scales.
>
> | F5-TTS small | Trained on LibriTTS WER | SIM | UTMOS | Trained on LJSpeech WER | SIM | UTMOS |
> |---------|----------:|-----|------|-------:|------|------|
> | Ground Truth  | 2.23 | 0.69 | 4.09  | 2.36  | 0.72  | 4.36 |
> | 100K updates | 29.5 | 0.53 | 3.78 | 5.64  | 0.72  | 4.17 |
> | 200K updates | 4.58 | 0.59 | 4.07 | 2.93  | 0.72  | 4.18 |
> | 300K updates | 2.71 | 0.60 | 4.11 | 3.26  | 0.71  | 4.12 |
> | 400K updates | 2.44 | 0.60 | 4.11 | 3.90  | 0.70  | 4.05 |
> | 500K updates | 2.20 | 0.60 | 4.10 | 4.68  | 0.70  | 3.99 |
> | 600K updates | 2.23 | 0.59 | 4.10 | 5.25  | 0.69  | 3.93 |
>
> The UTMOS with SIM-o implies promising performance training with a certain amount of high-quality data.
>
> Thanks again for the constructive feedback which also brings discoveries for us. We are always open to further questions and would be happy to discuss.

---

### Official Review · Reviewer_abUH · 2024-11-02

**Soundness:** 2
**Presentation:** 2
**Contribution:** 1
**Rating:** 3
**Confidence:** 5

**Summary:**

The paper introduces F5-TTS, a non-autoregressive text-to-speech (TTS) system utilizing flow matching with Diffusion Transformer (DiT). F5-TTS uses ConvNeXt to enhance text representation and introduces a Sway Sampling strategy to boost performance and efficiency compared to E2-TTS model.

**Strengths:**

The paper is clearly written and easy to follow, effectively outlining its research goal of addressing the slow convergence speed and low robustness of E2-TTS, a prior work in the emerging TTS paradigm. By increasing the capacity of text representation and improving inference sampling, it presents a clear approach to tackling these issues.

**Weaknesses:**

We expect a scientific paper rather than a technical report. Based on this expectation, the paper lacks the following content.

Originality and Significance:

- The paper's approach to addressing the issues of slow convergence and low robustness primarily involves augmenting the model with ConvNeXt for enhanced text representation. This raises concerns about whether simply adding ConvNeXt constitutes a sufficient contribution to the TTS research community. Fundamentally, it remains unclear why E2-TTS-like architecture is capable of learning some degree of alignment without cross-attention mechanisms. Is this alignment ability a consequence of scaling up the dataset?
- Beyond presenting favorable results, the paper lacks intuitive explanations for why these improvements occur. The paper does not provide performance comparisons with other architectures that naturally emerge as alternatives to ConvNeXt, such as Transformers [1], Conformers [2], or Mamba [3].
- The proposed Sway Sampling strategy appears to be a variation of the cosine schedule, which limits its novelty. The significance of inference sampling schedules in diffusion models is already well-known and has been extensively researched in prior works like DDPM [4], DDIM [5], and DPM-Solver [6].

Quality:

- There is a significant discrepancy between the scores in the original E2-TTS paper and the reproduced one. While it is understandable that reimplementation is necessary, this raises doubts about whether E2-TTS has been accurately reproduced.
- Also, only the reimplementation's metrics are compared for E2-TTS. It would be more consistent to compare either the reported metrics from all original papers or metrics from accurately reproduced implementations. Notably, when comparing with the original E2-TTS reported metrics, F5-TTS exhibits a higher WER.
- The use of Whisper-large-v3 for measuring WER differs from the evaluation models used in other papers. This raises concerns about the fairness and comparability of the results.
- Pronunciation errors still occur in demo samples—such as failing to pronounce "y'know" in the F5-TTS sample for "Disgust" in the Emotion section of the demo. Additionally, the prosody can become completely flat with abnormal input lengths, as seen in the 1.3x Speed sample for the first prompt in the "Speed Control" section of the E2-TTS demo page. It is challenging to ascertain how much these issues are mitigated compared to the original E2-TTS.
- For the above reasons, claiming state-of-the-art performance in the conclusion may be an overstatement.
- The paper mentions that it "simply estimates the duration based on the ratio," yet there appears to be a significant deviation when ground truth is provided. Given the performance gap, it might be beneficial to include additional modules to improve duration prediction, even if it compromises some efficiency. Using duration predictors from existing methods—such as summing phoneme durations to generate the total length, directly matching the total length—could potentially enhance performance.

Clarity:

- The paper lacks clear explanations for how several hyperparameters were determined. For example, why is the CFG scale set to 2? How were the ratios selected in the two-stage control of CFG? Referring to Figure 3, it seems that values of s < −0.8 might yield better metrics.
- In line 267, the paper mentions that having some gain when providing GT implies robustness. However, robustness is typically indicated when the difference in WER between using GT and not using GT is minimal. This reasoning needs further clarification.
- The term "re-ranking" used in line 461 is not adequately explained. The content described in lines 461-464 is difficult to interpret, and its significance and connection to the overall conclusions of the paper are not clear.
- In line 241, it seems that F5-TTS still exhibits significant length differences, similar to E2-TTS.
- There are several instances where proper citations are necessary to clarify references to existing work:
    - Lines 86-87: Is the paper referring to the monotonic alignment search method from Glow-TTS?
    - Line 317: Does this line relate to the cross-sentence task introduced in Voicebox?
    - Line 319: Is the paper referencing SIM-o from Voicebox?
    - Line 431: Does this pertain to the long skip connections used in DiTTo-TTS?

[1] Vaswani, A. (2017). Attention is all you need. *Advances in Neural Information Processing Systems*.

[2] Gulati, A., Qin, J., Chiu, C. C., Parmar, N., Zhang, Y., Yu, J., ... & Pang, R. (2020). Conformer: Convolution-augmented transformer for speech recognition. *arXiv preprint arXiv:2005.08100*.

[3] Gu, A., & Dao, T. (2023). Mamba: Linear-time sequence modeling with selective state spaces. *arXiv preprint arXiv:2312.00752*.

[4] Ho, J., Jain, A., & Abbeel, P. (2020). Denoising diffusion probabilistic models. *Advances in neural information processing systems*, *33*, 6840-6851.

[5] Song, J., Meng, C., & Ermon, S. (2020). Denoising diffusion implicit models. *arXiv preprint arXiv:2010.02502*.

[6] Lu, C., Zhou, Y., Bao, F., Chen, J., Li, C., & Zhu, J. (2022). Dpm-solver: A fast ode solver for diffusion probabilistic model sampling in around 10 steps. *Advances in Neural Information Processing Systems*, *35*, 5775-5787.

**Questions:**

- The phrase "Fairytaler Fakes Fluent" in the title does not clearly convey the content or main contributions of the paper.
- It is worth examining whether entanglement is a structural limitation in both E2-TTS and F5-TTS. To clarify, the paper could include an ablation study where, under the same configuration and data setting as F5-TTS, a modified model uses cross-attention for text conditioning instead of adding filler-padded text to speech. This would help determine if the existing entanglement issues are reduced by changing the conditioning method.
- The paper claims that the two-stage control of CFG aids text alignment, yet it does not experimentally demonstrate this. It would be beneficial to show, through experiments, what specific alignment issues arise with the original CFG training method and how the two-stage control addresses these problems.
- Line 357: Information about the Mean Opinion Score (MOS) evaluation process is insufficient. Clarifying whether a platform like MTurk was used, the number and qualifications of evaluators hired, and providing examples of evaluation scripts would improve transparency. Additionally, mentioning studies that used similar MOS evaluation standards could provide a basis for comparison.
- Line 426: The paper refers to the performance of "F5-TTS−Conv2Text," but no related data appears in any tables or figures. The phrase "fails to learn" suggests that this model entirely failed at learning alignment, which is puzzling given its similarity to E2-TTS in setup. An explanation of why this model fails, in contrast to E2-TTS, is needed to understand the effectiveness and limitations of ConvNeXt within the model.

typo

- While the captions of Tables 1 and 2 mention results *w/o* SS, the tables themselves do not include these values.
- Line 192 should explicitly designate the filler token as <F>.

---

> ### Author Response · Authors · 2024-11-20
> **Response to Reviewer abUH [1/3]**
>
> First of all, we want to thank the reviewer for carefully reading, spotting unclear points, and providing constructive feedback! Below we address the concerns mentioned in the review.
>
> > **W1 Orig. & Sign.**: 1. Whether sufficient contribution with the proposed simple refinement module; in-context-learning compared with cross-attention; and whether E2-TTS-like architecture has rigid demand to a large dataset. 2. Lack of model structure search with refinement module alternatives. 3. Novelty with Sway Sampling.
>
> 1. In-context-learning with Voicebox-like design has shown longstanding impressive performance, and E2 TTS has exactly the same structure but removing phoneme-level force alignment. We have first made efforts to diagnose the slow convergence and low robustness issues with the E2 TTS structure (Sec.5.1, Appendix B.2), and are not just adding to UNet-Transformer but equipping the adaptive layernorm DiT with ConvNeXt modest refinement. \
> \
> Compared to DiTTo-TTS (740M, English-only trained on 55K hours, cross-attention DiT with extra text encoder and codec joint-finetuning), our F5-TTS (336M, multilingual 100K hours, in-context-learning with simply padded character sequence) has -0.6 WER (-23\%), +0.043 SIM (+7\%), and -0.012 RTF (-7\%) on LibriSpeech test-clean (Table 1 and 8, Hubert-large-based model to transcribe). It is clear that our in-context-learning architecture has overall better performance, and a certain degradation of performance is expected if replacing modules in DiTTo-TTS with a simply padded character sequence, using in-the-wild data than audiobooks, and training with a smaller model size.\
> \
> We further include experiments in Appendix B.7 to address your concern about dataset scale. It is clear from evaluation results on small models trained on LibriTTS 585 hours and LJSpeech 24 hours, that our F5-TTS enables stable training to learn speech-text alignment (without grapheme-to-phoneme) with varying data amounts.
> | F5-TTS small | Trained on | LibriTTS | 585 hours |  Trained on |   LJSpeech | 24 hours  |
> |---|----:|:-----:|------|-------:|:-----:|------|
> |   | **WER** | **SIM** | **UTMOS** | **WER** | **SIM** | **UTMOS** |
> | Ground Truth  | 2.23 | 0.69 | 4.09 | 2.36  | 0.72  | 4.36 |
> | 100K updates | 29.5 | 0.53 | 3.78 | 5.64  | 0.72  | 4.17 |
> | 200K updates | 4.58 | 0.59 | 4.07 | 2.93  | 0.72  | 4.18 |
> | 300K updates | 2.71 | 0.60 | 4.11 | 3.26  | 0.71  | 4.12 |
> | 400K updates | 2.44 | 0.60 | 4.11 | 3.90  | 0.70  | 4.05 |
> | 500K updates | 2.20 | 0.60 | 4.10 | 4.68  | 0.70  | 3.99 |
> | 600K updates | 2.23 | 0.59 | 4.10 | 5.25  | 0.69  | 3.93 |
> (Update 2024/11/21 UTC-12, extra evaluation with UTMOS)
>
> 2. Our work focuses on providing a simple and effective modeling way for force-alignment-free NAR TTS and has put forward that a lightweight and modest refinement for padded character sequence is sufficient facilitating model to learn alignment, rather than exploring sweepingly best DiT or UNet-Transformer variant.
> 3. We have clearly stated in our main text that our Sway Sampling is an inference-time strategy while a commonly put cosine scheduler is applied for the training stage. While we acknowledge that we are not the first to emphasize the significance of inference sampling strategy, but as far as we know, there are not any previous strategies like our method that explicitly bias to smaller timesteps and can seamlessly incorporate with fixed-step ODE solvers that are closer to the works you have mentioned (just for reference, Voicebox and E2 TTS are using midpoint ODE solver).

---

> ### Author Response · Authors · 2024-11-20
> **Response to Reviewer abUH [2/3]**
>
> > **W2 Quality**: 1. Correctness of E2 TTS implementation. 2. Fairness and comparability of the results using Whisper-large-v3. 3. Overstatement to claim state-of-the-art performance 4. Demo samples issues. 5. Duration predictor.
>
> 1. We have had discussions with E2 TTS authors and confirmed the correctness of the implementation. Our base model is trained on an in-the-wild multilingual dataset which leverages Whisper-medium as the language identifier and to transcribe, and the comparisons are made with English-only models trained on much higher quality audiobook data, thus the reproduced E2 TTS has rational scores (WER 2.95 to ground truth 2.23, SIM-o 0.69 to 0.69).
> 2. We are encourage using a unified ASR though we are willing to provide results with the Hubert-large-based ASR model. Since the commonly used latter is finetuned on LibriSpeech, the scores are generally better while having similar relative values. To further ensure fairness, we use BigVGAN as the vocoder. As suggested, we compare with the original E2-TTS reported scores (denoted with asterisk) on LibriSpeech-PC (LS-PC) test-clean:
> | Model  | LS-PC test-clean WER | SIM |
> |-----|-----:|-----|
> | Ground Truth| 2.0*  | 0.695* |
> | Voicebox | 2.2* | 0.667* |
> | E2 TTS  | 2.0*  | 0.675* |
> | F5-TTS  | 1.81 | 0.67 |
>
> 3. In addition to the above results, we also updated more evaluation results in Appendix B.6, and we are encouraging comparisons among the same test sets and with the corresponding ground truth (otherwise distorted, e.g. our F5-TTS achieves a WER of 1.81, but we are only claiming that we are among the leading TTS systems in this case since the prompt pairs used are different).
>
> 4. We appreciate your careful listening to our demo samples. We are trying to present the results in the most rigorous way possible, so the numbers we report are the average of multiple runs and there are no cherry-picks for samples you mentioned. Thanks again for pointing out the shortcomings, which shows that there is a lot to explore in the field of TTS, and motivates us to continue to work hard.
>
> 5. We adopted the simple estimating way for simplicity, and the duration predictor will surely help.
>
> > **W3 Clarity**: 1. Lack of clear explanations for several hyperparameters' choices. 2. Reasoning with GT duration implies robustness. 3. Unclear explanation with E2 TTS consistent failures. 4. Other wording and referencing.
>
> 1. Due to the main text space limits and the expectation of the paper we all agreed, that we are not introducing solely hyperparameter searching ablations. Thanks for pointing out the Sway Sampling coefficient, which is updated in Appendix B.3. More aggressive Sway Sampling keeps improving performance and allows switching from 2nd-order midpoint to 1st-order Euler which brings significant speed up.
> | F5-TTS 16 NFE  | RTF | LS-PC test-clean WER | SIM | UTMOS | Seed-TTS test-en WER | SIM | UTMOS | Seed-TTS test-zh WER | SIM | UTMOS |
> |----|----|----:|----|----|-----:|----|----|-----:|----|----|
> | s=-1, Euler  | 0.15 | 2.53 | 0.66 | 3.88 | 1.89 | 0.67 | 3.76 | 1.74 | 0.75 | 2.96 |
> | s=-1, midpoint | 0.26 | 2.43 | 0.66 | 3.87 | 1.88 | 0.66 | 3.70 | 1.61 | 0.75 | 2.87 |
> | s=-0.8, Euler   | 0.15 | 2.82 | 0.65 | 3.73 | 2.14 | 0.65 | 3.70 | 2.28 | 0.72 | 2.74 |
> | s=-0.8, midpoint | 0.26| 2.58 | 0.65 | 3.86 | 1.86 | 0.65 | 3.68 | 1.70 | 0.73 | 2.83 |
> (Update 2024/11/21 UTC-12, extra evaluation with UTMOS)
>
> 2. Having some gain when providing GT does not imply robustness, but allowing "more" gain implies robustness in alignment which means F5-TTS can leverage better the truth information. At the same time, E2 TTS is relatively dull and insensitive.
>
> 3. "Re-ranking" is a commonly put term for retrieving an improved result with multiple sampling (e.g. conduct 5 times the generation and pick the best one based on metrics). The content you have mentioned is an indispensable part of E2 TTS's robustness issue analysis. The depiction of details of E2 TTS's "consistent failures" and ablation results with input condition that drops reference audio corroborate each other, and together have great significance and connection to verify that E2 TTS structure deeply entangles acoustic and semantic features.
>
> 4. Yes, F5-TTS still exhibits significant length differences, similar to E2-TTS, but "mitigated" with our refinement. \
> Thanks for pointing out the lack of reference to MAS with Glow-TTS, we have updated in main text. \
> The cross-sentence term might be phrased first in Voicebox but the task is not introduced by Voicebox (as far as we know, VALL-E is earlier than Voicebox). \
> SIM-o is put as "speaker Similarity between generated and the original target speeches", the same as "similarity against the original audio context" in Voicebox, or the reviewer is implying a lack of referencing to this "advocacy" from Voicebox (untypical, not referenced based on this term by NaturalSpeech 3, E2 TTS, etc.). \
> The long skip connection is from UNet, and DiTTo-TTS also borrows from it.

---

> ### Author Response · Authors · 2024-11-20
> **Response to Reviewer abUH [3/3]**
>
> > **Q1**: "Fairytaler Fakes Fluent" in the title does not clearly convey the content or main contributions of the paper.
>
> We all know Andersen's fairytale, "Fairytaler" bears our dream to advance TTS techniques that can read us a bedtime story. "Fakes" is a reminder for technical supervision. "Fluent" is for good prosody, and "Faithful" is for nice WER and SIM. "Flow matching" is the basic method.
>
> > **Q2**: Examining whether entanglement is a structural limitation in both E2-TTS and F5-TTS. To include an ablation study where a modified model uses cross-attention for text conditioning instead of adding filler-padded text to speech.
>
> As we have stated in W1-1 and W3-3, it is clear that our F5-TTS enables stable training and inference addressing E2 TTS robustness issues (with concrete analysis on entanglement of E2 TTS), and a cross-attention structure has no clear advantages compared to our design (F5-TTS has globally better performance to DiTTo-TTS which is trained on English-only data with a double model size and more complex design requirement extra training).
>
> | Model  | LibriSpeech test-clean WER | SIM |  RTF |  #Param. | #Data(hrs) |
> |-------|-----------------------:|----------------------|---------------|------------|:------------:|
> | DiTTo-TTS        | 2.56    | 0.627               | 0.162       | 740M      | 55K EN |
> | F5-TTS 16 NFE - w/ PC    | 1.96    | 0.67     | 0.15        | 336M      | 100K Multi.|
> | F5-TTS 16 NFE - w/o PC  | 2.34    | 0.67     | 0.15        | 336M      | 100K Multi.|
>
> > **Q3**: Lack of experiments to demonstrate the effectiveness of two-stage control of CFG. What specific alignment issues arise and how does the two-stage control address these problems?
>
> We appreciate your careful reading. Although not adequate for our main text to extend this experiment since not closely related to our main research problem and out of the scope of scientific research of "generative models", we are willing to share our informal assumption about it. \
> Since the two-stage control of CFG has a larger potential to let the model predict solely based on text, it intuitively enhances the model alignment ability for text modality which is harder to learn from a uniform observation (see evaluation results in the early stage of training). No "specific alignment issues" are outlined from our side, simply better performance.
>
> > **Q4**: Insufficient information about MOS. Mentioning studies that used similar MOS evaluation standards could provide a basis for comparison.
>
> We appreciate your advice. We have included Appendix C to provide more details of our subjective evaluations. Briefly, in both the CMOS and SMOS tests, 20 natives were invited to evaluate 30 rounds with randomly selected utterances for all test sets and models. To make a comparison as suggested, DiTTo-TTS received and reported 6 and 12 ratings for SMOS and CMOS, respectively, NaturalSpeech 3 invited 12 natives to judge 20 samples for CMOS and 10 samples for SMOS.
>
> > **Q5**: An explanation of why F5-TTS−Conv2Text (a pure DiT adaLN structure) failed.
>
> As the reviewer has mentioned, F5-TTS−Conv2Text is similar to E2-TTS in setup. With the same training configuration, when E2 TTS can produce intelligible speech, F5-TTS−Conv2Text still produces gibberish. Since the pure DiT adaLN structure is even more slimmed without any skip connections compared to E2 TTS (which already has issues with slow convergence and low robustness), it makes sense that F5-TTS−Conv2Text "fails to learn".
>
> > typo
>
> Thanks for mentioning. We have made corrections in the updated version.

---

> > ### Comment · Reviewer_abUH · 2024-11-23
> >
> > I appreciate the authors' detailed explanations and additional experiments; however, I remain unconvinced about the core contributions of the paper. Below, I outline the key unresolved issues that continue to raise concerns.
> >
> > > W1 Orig. & Sign.
> >
> > The paper identifies the slow convergence and low robustness issues in the E2 TTS structure as a contribution, framing it as a problem definition (PD). However, my question is whether the paper provides any insights into the causes of this PD and its solutions. To summarize my question in one sentence: What do the authors consider to be the research question of this study?
> >
> > When using adaLN DiT as the backbone, what evidence supports the conclusion that ConvNeXt is the best architecture for text refinement? For example, sequential data is often modeled using architectures such as Transformers, Conformers, or Mamba, which are well-suited for text representation. These are natural alternatives to consider for text refinement, and I believe comparative experiments with these models are essential to justify the choice of ConvNeXt.
> >
> > I understand that DiT was chosen as the backbone because it outperforms E2 TTS+Conv2Text. However, to draw such a conclusion, it is necessary to compare other plausible modeling options, such as replacing ConvNeXt with Transformers for text refinement or replacing the backbone with E2 TTS's UNet+Transformer or UNet. If such comparisons demonstrated that DiT with ConvNeXt consistently delivers better results, this would provide a valuable insight, such as showing that for text-speech alignment tasks, using DiT as the backbone and ConvNeXt as the convolution-based architecture for text refinement is preferable.
> >
> > My mention of data scale was in a similar vein. I was curious whether text-speech alignment, as modeled by E2 TTS, becomes feasible only with larger datasets. Given that TTS research has increasingly focused on large datasets in recent years, this modeling approach might work better in such contexts. However, based on the results included in the author response with smaller datasets, it seems the method works even with limited data, although there appears to be overfitting in LJSpeech. This suggests that success is not strictly tied to dataset size.
> >
> > Models like E2 TTS, which align text and speech without cross-attention, are unprecedented. While E2 TTS shows the potential of this approach, it does not provide insights into why this new modeling method works effectively. Researchers in the field may be surprised that alignment is achievable without cross-attention and eager to understand why. Therefore, for a study like F5-TTS, simply demonstrating improved performance over E2 TTS is an incremental contribution. Exploring the underlying reasons for the success of E2 TTS-like structures could offer critical insights that solidify the potential of this new modeling paradigm.
> >
> > > W2 Quality
> >
> > Thank you for the explanation and additional experiments. However, I still find the sample quality somewhat questionable compared to the reported table metrics. What are the confidence intervals (CIs) across multiple runs? Including performance improvements from duration prediction would also provide valuable insights for researchers.

---

> > > ### Comment · Reviewer_abUH · 2024-11-23
> > >
> > > > W3 Clarity
> > >
> > > Thank you for the detailed response and experimental results.
> > >
> > > 1. This information should be reflected in the main text (e.g., Figure 3). Could you also clarify how low the value of *s* can go (e.g., *s = -1.5*)? There is likely a sweet spot worth identifying.
> > > 2. It seems the term "robustness" is being used differently from its conventional meaning. Typically, robustness refers to maintaining strong performance even when inputs deviate from the ground truth distribution.
> > > 3. While I understand your intended meaning, the current phrasing in the paper suggests that F5-TTS physically reduces the difference in sequence input lengths. To convey your intention more clearly, it might be better to remove the term "unlike" and explicitly state that text refinement is designed to mitigate the impact of such length differences while they remain equal in magnitude.
> > > 4. While I agree with most of the explanations regarding citations, completely omitting them is problematic. If the authors are not the first to propose something, it should be properly cited:
> > >    - For cross-sentence tasks, both VALL-E and Voicebox should be cited.
> > >    - Regarding SIM-o, while NaturalSpeech 3 and E2 TTS did not cite it, that alone does not justify omitting citations here. Since their terminology is being used, both should be credited. Additionally, Voicebox deserves credit for initially clarifying the confusion around VALL-E’s similarity metrics and for introducing SIM-o and SIM-r.
> > >    - As the authors suggest, UNet should be cited. However, if the long skip connections used in the experiments connect only before and after passing through DiT blocks (instead of connecting inputs and outputs at every layer, as in UNet), then this is not a UNet design. In such cases, credit should be explicitly given to DiTTo-TTS, which first proposed this approach.
> > >
> > > > Questions
> > >
> > > - Q2: The paper defines entanglement as alignment failure when a speech prompt is given. I raised this question because I see this as a limitation of conditioning without cross-attention. For example, it would be interesting to investigate whether replacing the conditioning method with cross-attention in F5-TTS (while keeping the backbone DiT and ConvNeXt unchanged) reduces entanglement, even if the final performance does not improve. Based on Figure 1, replacing the conditioning mechanism with cross-attention appears to be a straightforward and intuitive modification.
> > > - Q3: The paper focuses on improving text-speech alignment, so it is unclear why the authors consider comparing performance with single-stage CFG beyond the scope of the study. Such comparisons seem essential for identifying the causes of improvement and align well with the paper’s stated objective.
> > > - Q5: If skip connections help stabilize training in the DiT structure but do not improve refinement with ConvNeXt (F5-TTS+LongSkip), what could be the underlying reason? Addressing this question is as critical as Q3, as understanding this phenomenon would provide valuable insights into the paper's overarching goal of improving text-speech alignment in E2 TTS-like architectures.

---

> ### Author Response · Authors · 2024-11-24
> **Replying to Response to Reviewer abUH [Cont. 1/2; W1,2]**
>
> We sincerely appreciate your feedback and are willing to have a point-to-point discussion.
>
> > W1 Orig. & Sign.
>
> - `What do the authors consider to be the research question (RQ) of this study?`
>   - **RQ1.1**: With an embarrassingly easy scheme and promising results, why E2 TTS has slow convergence and low robustness issues, what is the possible reason?
>   - **RQ1.2**: While keeping an easy scheme, is there a way to make training and inference faster, and stabler?
>   - **RQ2**: Is the new paradigm (no phoneme-level force-alignment) feasible in more general cases? e.g. on in-the-wild data, multiple languages, code-switching, etc.
>   - **RQ3**: Based on the observation both in training and inference, is there a better and explainable sampling strategy?
>
> - The Sway Sampling is one of our main points (**RQ3**). We have made a lot of efforts to address your concern (also R-mAVA Q4 and R-X8VX Q2) and would like to receive the reviewer's feedback.
>
> - `Is this alignment ability a consequence of scaling up the dataset?` seems addressed by our previous response while still listed (just to make sure).
>
> - `Fundamentally, it remains unclear why E2-TTS-like architecture is capable of learning some degree of alignment without cross-attention mechanisms` is the most concerned point and we would like to make it clear since closely related but different to our **RQ1.1**.
>   - *To make it clear*, Seed-TTS, DiTTo-TTS, and E2 TTS are concurrent works. Although the reviewer is giving superiority to cross-attention, we need to emphasize again that **in-context-learning (in Voicebox, etc.) is widely used across variable research fields** and **cross-attention is not the only way to achieve alignment and has no earliness on the new paradigm**. Thus we are not surprised that the alignment can be achieved and it is no less natural to use in-context learning than cross-attention.
>   - *We clarify the difference between the reviewer's feedback with our **RQ1.1**.* With a scientific perspective, we **located and disclosed the deep entanglement** of acoustic and semantic features with careful observation and experiments. Thus we propose an entanglement-mitigated adaLN DiT with **a lightweight and modest text refinement** module (**RQ1.2**).
>   - From the comparison between E2 TTS and DiTTo-TTS, a more complicated cross-attention structure with extra design has no significant advantages to a smaller and simpler in-context learning structure. Thus with **RQ1.2**, we come up with F5-TTS.
>   - While we acknowledge that a sweeping model structure searching will definitely help to further prove the superiority of adaLN DiT with ConvNeXt or find a better combination, **sufficient experiments are made** to confirm the **effectiveness** of our proposal to RQ1.2.
>
> - `Simply demonstrating improved performance over E2 TTS is an incremental contribution.`
>   - We personally think a `poor contribution` judgment for our work is not that fair.
>   - Apart from the abovementioned contributions (**RQ1 RQ3**) and given the background that many zero-shot large-scale models are not open-source and hard to evaluate, we not only made commendable efforts to compare with **many** current models (including **industrial** systems) and to demonstrate the **feasibility and high performance** on **in-the-wild data** with the new paradigm (**RQ2**).
>   - We also made huge efforts to give completely **transparent and reproducible** details of our **open-sourced system** presented in a paper `clearly written and easy to follow` as the reviewer has mentioned.
>
> > W2 Quality
>
> - Glad that we were able to address most of your concerns including `the correctness of E2 TTS reproduction`, `the fairness and comparability of the results`, and `claiming state-of-the-art performance in the conclusion may be an overstatement`.
>
> - We are willing to provide the result with CIs, under the reviewer's recommended evaluation setup:
> |F5-TTS|WER|SIM|
> |-|-|-|
> |16 NFE|1.958±0.077|0.667+0.005|
> |32 NFE|1.806±0.075|0.669+0.004|
>
> - It is clear that **WER 1.81 is fairly low**, and we may have some concerns that:
>   - If introducing a duration predictor would bring more significance as we have compared the results providing ground truth duration information, we may kindly ask what specific additional insights are expected as we are also delighted to see.
>   - Given a WER lower than ground truth, the result is probably distorted given the current ASR model's transcription, while the SIM is maintained the same as we have already mentioned in our paper.
>
> - `Find the sample quality somewhat questionable compared to the reported table metrics.`
>   - We have mentioned that **all code and checkpoints will be open-source**. We ensure that **all results are reproducible**.
>   - We are pleased that the reviewer has a high expectation of us (we have mentioned in the first response that no cherry-pick is made), and we would like to provide more information if necessary to prove an acceptable quality.

---

> ### Author Response · Authors · 2024-11-24
> **Replying to Response to Reviewer abUH [Cont. 2/2; W3, Q2,3,5]**
>
> > W3 Clarity
> 1. `Could you also clarify how low the value of s can go (e.g., s = -1.5)?` In section 3.2, we have mentioned $s ∈ [-1, 2/(π-2)]$.
> 2. We have removed the phrase "robustness" to avoid ambiguity. While we have the same observation with Seed-TTS that more "standardized" speech is easier for the ASR system to recognize, we believe in a more powerful ASR model in the future to better reflect zero-shot TTS models' nuanced differences in performance.
> 3. Many thanks for your advice. We have changed from 1 to 2:
>    1. ~And unlike the way of feeding the model with inputs of significant length difference (length with effective information) as E2TTS does, our design mitigates such gap.~
>    2. *Not directly feeding the model with inputs of significant length gap as E2 TTS does, the proposed text refinement mitigates the impact of using inputs with mismatched effective information lengths, despite equal physical length in magnitude as E2 TTS.*
> 4. We fully agree on the essentialness of giving proper credit to all works that have paved the way for our research.
>    1. VALL-E and Voicebox are cited for *cross-sentence* task.
>    2. Voicebox is credited for the clarification and advocacy of putting SIM-o.
>    3. Surely DiTTo-TTS is the first to explicitly propose "an additional long skip structure connecting the first to the last layer". We have properly referenced DiTTo-TTS with the method description, ensuring no ambiguity with the long skip connections proposed by U-Net.
>
> > Q2:
> > 1. Whether entanglement is a structural limitation in both E2-TTS and F5-TTS.
> > 2. Using cross-attention for text conditioning instead of adding filler-padded text to speech seems a straightforward and intuitive modification to reduce entanglement.
>
> - The first part of Q2 is unchanged and we have already made a detailed response. We would appreciate it if the reviewer could clarify whether the comprehensive comparisons we made can confirm that F5-TTS is liberated from the entanglement (**RQ1.2**) rather than suffers together with E2 TTS.
> - The second part of Q2 seems advanced with our text refinement method explicitly involved. From our perspective, DiTTo-TTS has conducted profound experimental analysis with model structure, in which the text encoder and probably joint finetuned codec play relatively important roles. Based on the experiences of DiTTo-TTS, it is hard to say our lightweight text refinement could replace the original text encoder and codec to achieve better performance with cross-attention than DiTTo-TTS. Moreover, in **Response to W1 Orig. & Sign.**, we hope our point regarding the effectiveness and naturalness of using the in-context learning pattern has been clearly conveyed.
>
> > Q3: Why not compare the performance of two-stage control of CFG with single-stage one which seems essential for identifying the causes of improvement?
>
> - We acknowledge that it was our oversight. We were assuming that the Voicebox's setting of 0.3 for $p_{drop}$ and a $p_{uncond}$ of 0.2 is commonly put as well as the 70% to 100% random mask. Ensuring closest reproduction, the 0.3/0.2 is the same with E2 TTS (following Voicebox).
> - We have updated with referencing Voicebox to split between the configuration setting and our new terminology of two-stage control of CFG. Even though it was for misunderstanding and mainly our oversight, we would like to thank the reviewer again for careful reading and interest in it.
> - We hope it is clear now that we are not avoiding answering this question and are definitely not claiming to propose this setting. For the abovementioned reasons, the two-stage CFG does not additionally benefit F5-TTS compared to Voicebox/E2 TTS.
>
> > Q5:
> > Review-1: Need to explain why a pure DiT adaLN failed in contrast to E2-TTS to understand the effectiveness and limitations of ConvNeXt.
> > Review-2: If skip connections help stabilize training in DiT but do not improve refinement with ConvNeXt (F5-TTS+LongSkip), what could be the underlying reason? ... would provide valuable insights for improving text-speech alignment in E2 TTS-like architectures.
>
> - We hope the R1-Q5 has been addressed with our first response.
> - R2-Q5 is a different but similar question regarding the model design, consistent with R1-Q5, concerning the effectiveness of ConvNeXt for improvement of text-speech alignment.
>   - We assume that the reviewer is referring to a singular form skip connection as in F5-TTS+LongSkip. In the ablation study, this structure shows inferior performance to F5-TTS and is not more stable in training. From the comparison of F5-TTS, F5-TTS+LongSkip, and analysis in our response to R1-Q5, a conclusion that the text refinement ConvNeXt to DiT helps stabilize training makes more sense.
>   - One of our main points in this paper is with **RQ1.2** while the analysis of how ConvNeXt is refining padded text input (e.g. analyze the output of ConvNeXt, probe E2 TTS and F5-TTS transformer first layers) serves as our future research question.

---

> > ### Comment · Reviewer_abUH · 2024-11-25
> >
> > Thank you for the authors' response. It has helped clarify the areas where our perspectives differ. I have summarized my remaining concerns and feedback below.
> >
> > > W1 Orig. & Sign.
> >
> > Thank you for organizing and clarifying the RQs. This has made it much clearer what the authors aim to address, aside from the points noted below, and I believe the paper has improved in terms of clarity, even if it doesn't fully address my concerns.
> >
> > - In **RQ1.1**, the "why" and "what" appear to be asking the same question. Was this intentional? If there is a distinction, could you please provide clarification?
> > - If the RQs are considered critical to the paper, especially **RQ2**, which appears dispersed throughout the text, restructuring the manuscript to make these RQs more prominent would improve clarity.
> >
> > Through the clarification of the RQs, I feel more certain that the paper does not fully address them and that its contributions are incremental for a scientific paper. I will elaborate on this in my responses to the authors' explanations.
> >
> > I acknowledge that my repeated references to cross-attention may have given the impression of "giving superiority to cross-attention." However, I want to clarify that I do not favor, nor is there a need to favor, any specific approach. Instead, I aim to point out that the authors, relying on insufficiently substantiated insights, appear to be attributing undue superiority to the "in-context-learning" approach (as they describe it) used in E2 TTS.
> > - First, it seems the authors are misusing the term "in-context-learning problem" by tying it to the conditioning method. In Voicebox, "in-context-learning" refers to "...a guided in-context learning problem, where audio style is inferred from the audio context and textual content is specified through transcript," which involves masked prediction based on audio prompts and textual content. This definition is independent of the conditioning method. Therefore, it would be reasonable to consider that both E2 TTS and DiTTo address the in-context-learning problem.
> > - Traditional text conditioning methods can be broadly categorized into two approaches: using cross-attention, as in AR and Non-AR models, or relying on phoneme-level alignment (duration) without cross-attention (like Voicebox). E2 TTS proposes a novel method that differs from both, conditioning solely on padded text without cross-attention or phoneme-level alignment. The fact that text-speech alignment is achieved without explicitly modeling or utilizing alignment is undoubtedly surprising to researchers who considered traditional methods valid, and this is likely the primary reason why E2 TTS is regarded as a valuable contribution.
> > - The issue of "the deep entanglement of acoustic and semantic features," which the authors highlight as central to RQ1.1, does not appear to have been addressed in previous studies on the in-context-learning problem and seems to have been first identified as an issue in the E2 TTS model. Furthermore, the authors show that this issue can be mitigated through text refinement, indicating that it is likely dependent on the conditioning method. A natural follow-up question, then, is whether this issue specifically arises due to the absence of cross-attention and phoneme-level alignment in the conditioning method. Therefore, I point out the lack of discussion or experimental (or theoretical) evidence in the paper to explain the root cause of this issue.
> > - Additionally, the experiments and questions I previously suggested (e.g., for RQ1.1, why must the "slow convergence and low robustness issues" be addressed specifically with the DiT + ConvNeXt combination? Could ConvNeXt be replaced with another option? Why does the UNet + ConvNeXt combination fail? Or for Q5, why did F5-TTS−Conv2Text fail?) are all essential to substantiate the claim that the RQs outlined by the authors can be addressed through the DiT + ConvNeXt combination.
> > - Without addressing these questions, it is difficult to agree with the statement that "sufficient experiments are made to confirm the effectiveness of our proposal to RQ1.2." While the results from "the comparison between E2 TTS and DiTTo-TTS" demonstrate that the proposed approach can solve the given problem, they merely show that the approach works in practice. They do not provide insights into the underlying causes, such as why "the deep entanglement of acoustic and semantic features" arises, which limits the contribution beyond empirical findings.
> >
> > The authors' response ("F5-TTS enables stable training ... with varying data amounts.") confirms that the proposed method performs well even with smaller datasets. This addresses my original question—whether this new text conditioning method works only because of larger datasets—with a clear "no." However, the intent behind my original question was to examine the contribution more closely, and I provided further explanation in my previous response for clarification.

---

> > > ### Comment · Reviewer_abUH · 2024-11-25
> > >
> > > > W2 Quality
> > >
> > > - Thank you for the clarification. As you mentioned, since "the correctness of E2 TTS reproduction" has been addressed, "the fairness and comparability of the results" are also resolved.
> > > - However, based on the shared CI results, it seems that the quality mostly aligns with demo sample levels. From this perspective, I feel that "claiming state-of-the-art performance in the conclusion may be an overstatement," as it does not fully align with the objective metrics. That said, the discussion on the validity of the final numbers is of lower importance compared to the discussion on contributions mentioned earlier, and I would like to revisit this in the future if necessary, alongside open-source reproducibility.
> > > - The insights that can be derived from the duration predictor are quite straightforward: when the gap between the GT and the current ratio-based estimation is significant, the duration prediction allows us to measure how much this gap can be reduced.
> > >
> > > > W3 Clarity
> > >
> > > - I realize my question may have been unclear. I understand that experiments were conducted up to \(s = -1\), and performance improved compared to \(s = -0.8\). I was curious whether performance might improve further at \(s = -1.5\). My inquiry about the trade-off relates to the likely limit on how small the \(s\) value can go, and whether performance continues to improve as \(s\) decreases further.
> > > - Thank you for clarifying the points about robustness, significant length differences, and citation. Your explanation has made these aspects much clearer.
> > >
> > > > Q2
> > >
> > > I have included discussions in "W1 Orig. & Sign."
> > >
> > > > Q3
> > >
> > > It is difficult to immediately identify the changes in the revised script, but I did notice that the term "two-stage" has been removed. Based on the authors' response, it seems the issue was that the original phrasing made it appear as though a new method was being proposed. My understanding is that the explanation has now been updated to cite the existing method (presumably Voicebox's) instead. Could you confirm if this interpretation is correct?
> > >
> > > > Q5
> > >
> > > R1-Q5 remains unresolved, as I have detailed in "W1 Orig. & Sign." While I agree with the demonstrated result that "ConvNeXt to DiT helps stabilize training," the question I raised from the very first feedback—why ConvNeXt is the necessary choice over other options—remains unanswered. I believe this is not merely a "future research question" but rather a critical point required to substantiate the current claims, as I have discussed in detail in "W1 Orig. & Sign."
> > >
> > >
> > > In summary, I want to explicitly acknowledge that I have recognized the improved results and several key contributions of this work from my initial response. However, I view this paper as leaning more towards a technical report, and in this context, I consider the research contribution to TTS to be incremental. While I understand that the authors may disagree with this assessment and welcome further discussion on the matter, my evaluation is unlikely to change unless my questions are addressed. Therefore, I will maintain my current score.

---

> > > ### Author Response · Authors · 2024-11-25
> > > **Response 3 to Reviewer abUH [1/2]**
> > >
> > > We sincerely appreciate your feedback and value your efforts to help improve our work.
> > >
> > > > W1 Orig. & Sign.
> > >
> > > 1. Most importantly, we have mentioned and emphasized one of our main contributions is to propose **Sway Sampling** (**RQ3**) while the reviewer has cast a veil over it twice. Thus we think the `poor contribution` judgement is based on an incomprehensive review.
> > >
> > > 2. Since the reviewer insists on the earliness of cross-attention for text conditioning without phoneme-level duration but does not provide any specific reference in this field (the reviewer has made explicit classification with modeling pattern w/ w/o phoneme-level duration), we would like to receive a rigorous and comprehensive discussion that our efforts to this are valued.
> > >
> > > 3. **Here we summarize again our contributions** (`Contribution: 1: poor` judged by the reviewer):
> > >     - We draw attention to the slow convergence and low robustness issues of E2 TTS under the new paradigm without phoneme-level alignment and disclose the underlying deep entanglement of acoustic and semantic features with careful experiments.
> > >     - We propose to address with F5-TTS (DiT with adaptive layernorm incorporating a lightweight joint-learned text refinement ConvNeXT) and prove its effectiveness and efficiency with commendable evaluations.
> > >     - We propose an inference-time sampling strategy that can be seamlessly incorporated and comprehensively improves the performance of existing flow-matching models without retraining.
> > >     - We explore the feasibility of F5-TTS on large-scale in-the-wild data (and also for limited data scale e.g. 585 hours or 24 hours data, as suggested by R-7LV6) with impressive zero-shot multilingual code-switching ability.
> > >     - We ensure all transparency and reproducibility of our open-source system (releasing all code and checkpoints).
> > >
> > > > W2 Quality
> > >
> > > 1. `However, based on the shared CI results, it seems that the quality mostly aligns with demo sample levels. From this perspective, I feel that "claiming state-of-the-art performance in the conclusion may be an overstatement," as it does not fully align with the objective metrics.` The feedback is not very clear, for example, what CI results are expected? As far as we know, a WER of 1.806%±0.075% is the lowest compared to other models, and a SIM of 0.669+0.004 is better than more than half of the leading TTS systems. A clear and constructive comment would be appreciated.
> > > 2. `The discussion on the validity of the final numbers is of lower importance compared to the discussion on contributions`, `I would like to revisit this in the future if necessary`, as we are making all efforts to follow the code-of-conduct as authors, we hope that the reviewer can keep on taking the responsibility. Also, Soundness, Presentation and Contribution are three main criteria that are all important.
> > > 3. **Here we summarize the discussion of quality**
> > >     - R-mAVA: `The model performance is good and the audio sample is impressive.`, R-7LV6: `The model demonstrates respectable speaker similarity, audio quality, and pronunciation accuracy in English and Chinese benchmarks.`, R-X8VX `3: good`, and the reviewer abUH `Beyond presenting favorable results`.
> > >     - We are happy that the fairness and correctness concerns of our evaluation were addressed, and there are still some concerns about our provided results (under the setup suggested by the reviewer) which are somehow vague and might need some clarification from the reviewer.
> > >     - Although the reviewer insists on introducing an additional duration predictor to make more comparisons and disregards the limit of the capacity of current ASR models, we personally do not feel the need given our results are comparable or even better than ground truth on test sets.

---

> ### Author Response · Authors · 2024-11-25
> **Response 3 to Reviewer abUH [2/2]**
>
> > W3 Clarity
>
> 1. As we have mentioned in the earlier response: the Sway Sampling function `is monotonic with coefficient s∈[-1, 2/(π-2)]`, it is obvious mathematically that going out of the monotone interval of the given function is mapping a certain uniformly sampled `u` to negative values which are invalid for conditional flow matching. We hope it is clearly enough now for the reviewer concerning about the performance of a lower value `s=-1.5`.
>
> 2. We are always willing to address your concerns and thanks again for pointing out unclear points.
>
> > Q3: Based on the authors' response, it seems the issue was that the original phrasing made it appear as though a new method was being proposed. My understanding is that the explanation has now been updated to cite the existing method (presumably Voicebox's) instead. Could you confirm if this interpretation is correct?
>
> As we have mentioned previously, we assumed that the setup of Voicebox is widely known, and the reviewer has not mentioned other hyperparameters in the same paragraph. The update is to explicitly cite Voicebox for the hyperparameter setup sentence before our new terminology in a one-sentence assumption. Voicebox takes $p_{drop}$ and 70% to 100% random mask together for infilling setup, and $p_{uncond}$ for CFG, while we describe in our paper $p_{drop}$ and $p_{uncond}$ with the new terminology "two-stage control of CFG". All these things belong to the "Experimental Setup" section where a new method is not introduced here.
>
> > Q5
>
> The future research questions we mentioned in the previous response are quite different from the ablation of the model structure suggested by the reviewer. We believe the future research questions we mentioned are scientific and are to explore the essence.
>
> \
>
> Thanks again for the reviewer's rigorous comments. Although in some areas our perspectives differ, we fully respect your ideas.
>
> As we have mentioned earlier, some other areas have not been fully discussed but are crucial for both our work and ensuring comprehensiveness of judgment. We would be delighted to receive the reviewer's feedback.

---

> ### Comment · Reviewer_abUH · 2024-11-26
>
> Thank you for the authors' kind and detailed responses. Below is my feedback.
>
> > W1 Orig. & Sign.
>
> 1. In my previous response, I omitted a paragraph while attempting to meet the character limit. This was clearly an error on my part, and I apologize if it caused the authors to perceive my review as incomprehensive. I want to emphasize that I had no intention of deliberately dismissing contributions that deserve recognition. My intent was to clarify that I recognize and agree with some of the contributions highlighted by the authors. To this end, I wrapped up my previous response with the statement, "I want to explicitly acknowledge that I have recognized the improved results and several key contributions of this work from my initial response." I hope the authors will take this clarification into account when interpreting my previous feedback.
>
>     - I acknowledge that the proposed sampling method is well-explained and contributes to the final performance of F5-TTS, even though, as I noted in my initial review, the importance of sampling methods in diffusion is not a novel insight. However, since it addresses only one of the research questions, I am uncertain whether it is sufficient to establish the paper’s overall contribution. I also recognize the significance of open-source challenges within the TTS research community and appreciate the authors' efforts toward open-source contributions. Furthermore, as highlighted in RQ2, I value the authors' work in addressing underexplored areas such as in-the-wild data and code-switching, though multiple languages have already been investigated in works like Voicebox. Nonetheless, these contributions are of lower importance and priority compared to my concerns regarding the paper’s main contribution.
>
>     At the time I submitted my last response, the announcement about the extension of the discussion period had not yet been made, and only limited time remained. I kindly ask for your understanding, as I was rushing to prepare my response, knowing that continuing the discussion with the authors within the available time was the best way I could assist them.
>
> 2. I followed the classification initially provided by the authors to rephrase and elaborate on my response, so I did not consider a citation necessary. However, it is now clear that the authors did not fully acknowledge the pre-existing nature of cross-attention, and I believe this requires correction.
>
>     - **Cross-attention:** Broadly speaking, cross-attention has been a foundational component in conventional encoder-decoder TTS models, starting with the strong baseline Tacotron series [1, 2], followed by Transformer TTS [3]. More recent works include SPEAR-TTS [4], Simple-TTS [5], E3 TTS [6], CLaM-TTS [7], and T5-TTS [8]. Among concurrent works, Seed-TTS and DiTTo-TTS, mentioned by the authors, also utilize cross-attention.
>     - **Phoneme-level alignment:** This is arguably the most conventional approach, first introduced by the FastSpeech series [9, 10], which used duration predictors based on forced alignment obtained from encoder-decoder teachers or tools like the Montreal Forced Aligner. Many subsequent works have adopted this monotonic alignment modeling for TTS tasks, including Glow-TTS [11] and its numerous variants such as VITS [12] and Your-TTS [13]. Additionally, large-scale architectures like Voicebox and the NaturalSpeech series [14, 15, 16] have continued to build on this approach.
>
>     On the other hand, I was surprised to learn that the authors were unfamiliar with this classification. Given that the authors' work makes its main claims based on text conditioning (and speech-text alignment) with the distinctive E2 TTS framework, this is foundational knowledge that should have been addressed. It appears that the paper lacks a dedicated related work section. I strongly suggest including a separate paragraph to summarize text conditioning methods, incorporating the models I mentioned, to provide proper context and background for the proposed approach.
>
> 3. Thank you for providing a summary once again. The points you raised are consistent with the claims already made in your responses, without any notable differences. I will restate the key aspects based on my previous feedback:
>
>     - As mentioned earlier, while I acknowledge that the authors "draw attention" to and "disclose" the issue, the analysis of its causes (e.g., the deep entanglement of acoustic and semantic features) is still insufficient.
>     - Similarly, although the final performance comparisons and ablations aim to "prove" the effectiveness of the current approach, the justification lacks deeper insights into its validity. Experiments exploring the underlying causes, which are critical, remain insufficient.
>
>     There is a lack of insight into the question, "What exactly does text refinement entail?" For further clarification on examples and remaining points, please refer to my response to point 1 above and my previous feedback.

---

> > ### Comment · Reviewer_abUH · 2024-11-26
> >
> > > W2 Quality
> >
> > 1. Given the limited number of demo samples, I aimed to determine, through CI, whether the issues I observed in randomly selected samples (as noted in my initial review) were also prevalent in others. Since it is impractical to review all samples individually, CI provided a way to evaluate the variance. The authors' experiments confirmed low variance in the CI results, leading me to conclude that the samples are generally of similar quality. I wanted to highlight that if most audio samples exhibit these issues, there seems to be a noticeable disconnect between the metrics and the perceived quality.
> >
> > 2. However, I view this discussion as lower in priority compared to the core issue of the contribution, which remains central to our dialogue. Similar to the quality-related points mentioned in item 1, I believe further progress on this matter would require listening to all audio samples or running the model directly for verification, which is impractical at this stage. Therefore, I suggested addressing it later if necessary. I want to emphasize again that I consider the issues related to the contribution to be more significant and that my evaluation reflects concerns about both soundness and presentation.
> >
> >     I acknowledge the authors' efforts to address my concerns. However, I want to emphasize that I have also dedicated significant time and effort to providing constructive and respectful feedback aimed at improving this work, regardless of the evaluation score. I remain fully committed to fostering meaningful discussions and believe I am fulfilling my responsibilities as a reviewer with diligence and accountability.
> >
> > 3. I respect the evaluations of other reviewers; however, my assessment is independent of theirs. My judgment is based on the high quality bar I adhere to, as the authors have acknowledged, and simply reflects the differing standards held by each reviewer.
> >
> >     In my initial review, I outlined the issues I observed in the audio samples, and the authors responded by acknowledging my concerns and clarifying that no cherry-picking was involved. As explained in point 1, my examination of the CI aimed to assess the relative quality of the remaining samples and confirm the possibility of similar issues. I believe this evaluation process has been clearly explained, and I am uncertain where the authors perceive ambiguity or what specific aspects they feel require further clarification in their inquiry.
> >
> >     As I have emphasized repeatedly, the purpose of the duration predictor experiments is not to improve the final results but to strengthen the analysis from a research perspective.
> >
> > > W3 Clarity, Q3, Q5
> >
> > Thank you for the detailed explanation. It has clarified the questions I had regarding the authors' previous response.

---

> > > ### Author Response · Authors · 2024-11-26
> > > **Response 4 to Reviewer abUH [2/2]**
> > >
> > > > W2 Quality
> > >
> > > Thanks for the reviewer's clarification of previous comments. We confirm that we share the same understanding that the low variance of our CI results shows that our demo samples are generally of similar quality. Thus we hope the reviewer could find our following perspectives to quality validation aligned.
> > >
> > > 1. `Given the limited number of demo samples, I aimed to determine, through CI, whether the issues I observed in randomly selected samples were also prevalent in others. ... I wanted to highlight that if most audio samples exhibit these issues, there seems to be a noticeable disconnect between the metrics and the perceived quality.`
> > >    - We fully value the reviewer's time and respect the decision to randomly select samples for observation (we are not against it since we do it ourselves sometimes). However, as we have mentioned, no cherry-picking is involved for our 68 samples on the demo page. There could be flaws in the picked ones since our model is far from perfect.
> > >    - Moreover, the lowest WER of 1.806%±0.075% and a fairly good SIM of 0.669+0.004 are evaluated based on 1127 samples (and we also provide scores evaluated on other test sets with 1088 and 2020 samples)
> > >    - In addition to objective metrics, our model has shown impressive results on subjective metrics CMOS and SMOS, 20 natives were invited to evaluate 30 rounds with randomly selected utterances for all test sets and models, to be clear, 30 rounds respectively (comparison with other studies that used similar MOS evaluation standards are included as the reviewer suggested, to confirm the validity of setup).
> > >    - Thus we personally think that drawing a conclusion that our results are questionable based on few samples and inferred to the general performance may not be that rigorous and comprehensive. On the other hand, a low variance indicates that the majority of samples are good. The latter causality makes more sense to us.
> > >
> > > 2. We understand that the reviewer is prioritizing Contribution more, and we sincerely appreciate the reviewer's diligence and accountability. Our response to your concern here is the same as the abovementioned, and we are pleased that the reviewer could also make direct verification in the future as we are confident about our model's performance.
> > >
> > > 3. We acknowledge that the reviewer has high expectations for our work, which makes us grateful. Generally speaking, we are making comparisons to baseline models rather than an ideal perfect one.
> > >
> > > 4. `As I have emphasized repeatedly, the purpose of the duration predictor experiments is not to improve the final results but to strengthen the analysis from a research perspective.`
> > >    - We believe the reviewer is very familiar with speech area works. As in previous responses, we would like to discuss with the reviewer whether a distortion of results would occur leveraging current ASR models and given our model's current performance (better than GT in some test sets).
> > >    - We agreed with the reviewer that including more experiments with duration predictors will surely provide insights, but we kind of feel that it deviates from our RQs. If the reviewer is expecting for ablation study with "more" accurate duration information, we have already compared with results providing ground truth duration information.
> > >    - Since the reviewer has a certain concern with our objective results, if the experiments are carried out assuming the ablation with introducing duration predictor aligns with our RQs and is free of distortion from ASR models, we think the validity might also be doubted.
> > >
> > > > W3 Clarity
> > >
> > > Glad that we were able to address your concerns. We are always willing to ensure a clear presentation of our paper with the least ambiguity and that no proper reference is omitted.
> > >
> > > \
> > >
> > > We would be delighted to receive the reviewer's feedback, especially looking forward to your possible concerns about RQ3 and our Sway Sampling strategy which is not fully covered in the previous discussion.

---

> > ### Author Response · Authors · 2024-11-26
> > **Response 4 to Reviewer abUH [1/2]**
> >
> > We sincerely appreciate the reviewer's feedback and are delighted to bring into the discussion areas where ideas have not been sufficiently exchanged.
> >
> > > W1 Orig. & Sign.
> >
> > 1. `The importance of sampling methods in diffusion is not a novel insight`
> >    - From the first and previous responses, we found that our proposed Sway Sampling is largely underestimated and we hope to make it clear if the reviewer has any concerns.
> >    - Firstly, our method (detailed introduced in our paper) is different from the training-time scheduler or the inference-time ODE solvers in the field, of great novelty (as far as we know, there are not any previous strategies like our method that explicitly bias to smaller timesteps) and significance (points 1.3, 1.4).
> >    - The Sway Sampling has strong interpretability (already elaborated in our paper and previous responses) and is improving comprehensively the performance of not only F5-TTS but also other CFM-based models. In addition to TTS models, the proposed method can also be easily applied to other research fields, e.g. for a CFM-based Video Generation model.
> >    - Moreover, the inference-time Sway Sampling can be seamlessly used with existing CFM-based models without the need for retraining, which is of great usability. It can also incorporate various ODE solvers during inference, e.g. Euler, midpoint (used by Voicebox/E2 TTS), Heun, etc.
> >
> >     We appreciate the reviewer for acknowledging many of our contributions regarding RQ2 & RQ3 and we fully respect the reviewer's decision holding concerns about RQ1.
> >
> > 2. `I followed the classification initially provided by the authors to rephrase and elaborate on my response, so I did not consider a citation necessary.`, `On the other hand, I was surprised to learn that the authors were unfamiliar with this classification.`.
> >    - We personally think our previous response is not worth being interpreted as `unfamiliar with this classification`.
> >    - We have clearly stated in our paper that AR models `perform implicit duration modeling` and `Unlike AR-based models, the alignment modeling between input text and synthesized speech is crucial and challenging for NAR-based models`. We cited and introduced many related works and focused on NAR, which we assumed contextualized in our discussion.
> >    - Our previous response `Since the reviewer insists on the earliness of cross-attention for text conditioning without phoneme-level duration but does not provide any specific reference in this field (the reviewer has made explicit classification with modeling pattern w/ w/o phoneme-level duration), we would like to receive a rigorous and comprehensive discussion that our efforts to this are valued.` is purely based on the fact that the reviewer has referenced `Voicebox` for models `relying on phoneme-level alignment (duration) without cross-attention`, but no reference for models `using cross-attention, as in AR and Non-AR models`, and the reviewer's consistent insistence of the earliness of cross-attention without phoneme-level alignment.
> >    - To further clarify the previous point, we were expecting some referenced NAR models that have acceptable performance using cross-attention without phoneme-level alignment earlier than E2 TTS (but we would like to express our sincere thanks to the reviewer for providing a detailed reference list). We know that Simple-TTS and E3 TTS have made some explorations, but do they follow the assumption of `Researchers in the field may be surprised that alignment is achievable without cross-attention and eager to understand why.` and `Fundamentally, it remains unclear why E2-TTS-like architecture is capable of learning some degree of alignment without cross-attention mechanisms` in the opinion of the reviewer? To be more specific, under the new paradigm, does the performance of Simple-TTS and E3 TTS meet the bar of the reviewer to serve as earlier `approaches work in practice` as mentioned for DiTTo-TTS and E2 TTS, or an E2 TTS-like design is never expected previously to have similar performance as the former two?
> >    - Minor typo of the reviewer: we have not mentioned Seed-TTS is utilizing cross-attention (plural form `utilize` used by the reviewer). Seed-TTS authors have not elaborated on their model structure).
> >
> >    While making sure that our ideas are accurately conveyed, we have tried our best to change the wording to be the most subtle and gentle. We always hope that our discussion is to make improvements and we really appreciate your efforts.
> >
> > 3. Thanks for the reviewer's acknowledgment of our attention-drawing (to researchers) disclosure of issues in E2 TTS. The summary in our previous response is to ensure the consistency of reading for everyone since we had detailed and thorough discussions.

---

> ### Comment · Reviewer_abUH · 2024-11-26
>
> > References
>
> [1] Wang, Y., Skerry-Ryan, R. J., Stanton, D., Wu, Y., Weiss, R. J., Jaitly, N., ... & Saurous, R. A. (2017). Tacotron: Towards end-to-end speech synthesis. arXiv preprint arXiv:1703.10135.
>
> [2] Shen, J., Pang, R., Weiss, R. J., Schuster, M., Jaitly, N., Yang, Z., ... & Wu, Y. (2018, April). Natural tts synthesis by conditioning wavenet on mel spectrogram predictions. In 2018 IEEE international conference on acoustics, speech and signal processing (ICASSP) (pp. 4779-4783). IEEE.
>
> [3] Li, N., Liu, S., Liu, Y., Zhao, S., & Liu, M. (2019, July). Neural speech synthesis with transformer network. In Proceedings of the AAAI conference on artificial intelligence (Vol. 33, No. 01, pp. 6706-6713).
>
> [4] Kharitonov, E., Vincent, D., Borsos, Z., Marinier, R., Girgin, S., Pietquin, O., ... & Zeghidour, N. (2023). Speak, read and prompt: High-fidelity text-to-speech with minimal supervision. Transactions of the Association for Computational Linguistics, 11, 1703-1718.
>
> [5] Lovelace, J., Ray, S., Kim, K., Weinberger, K. Q., & Wu, F. (2023). Simple-TTS: End-to-End Text-to-Speech Synthesis with Latent Diffusion.
>
> [6] Gao, Y., Morioka, N., Zhang, Y., & Chen, N. (2023, December). E3 tts: Easy end-to-end diffusion-based text to speech. In 2023 IEEE Automatic Speech Recognition and Understanding Workshop (ASRU) (pp. 1-8). IEEE.
>
> [7] Kim, J., Lee, K., Chung, S., & Cho, J. (2024). CLaM-TTS: Improving Neural Codec Language Model for Zero-Shot Text-to-Speech. arXiv preprint arXiv:2404.02781.
>
> [8] Neekhara, P., Hussain, S., Ghosh, S., Li, J., Valle, R., Badlani, R., & Ginsburg, B. (2024). Improving robustness of llm-based speech synthesis by learning monotonic alignment. arXiv preprint arXiv:2406.17957.
>
> [9] Ren, Y., Ruan, Y., Tan, X., Qin, T., Zhao, S., Zhao, Z., & Liu, T. Y. (2019). Fastspeech: Fast, robust and controllable text to speech. Advances in neural information processing systems, 32.
>
> [10] Ren, Y., Hu, C., Tan, X., Qin, T., Zhao, S., Zhao, Z., & Liu, T. Y. (2020). Fastspeech 2: Fast and high-quality end-to-end text to speech. arXiv preprint arXiv:2006.04558.
>
> [11] Kim, J., Kim, S., Kong, J., & Yoon, S. (2020). Glow-tts: A generative flow for text-to-speech via monotonic alignment search. Advances in Neural Information Processing Systems, 33, 8067-8077.
>
> [12] Kim, J., Kong, J., & Son, J. (2021, July). Conditional variational autoencoder with adversarial learning for end-to-end text-to-speech. In International Conference on Machine Learning (pp. 5530-5540). PMLR.
>
> [13] Casanova, E., Weber, J., Shulby, C. D., Junior, A. C., Gölge, E., & Ponti, M. A. (2022, June). Yourtts: Towards zero-shot multi-speaker tts and zero-shot voice conversion for everyone. In International Conference on Machine Learning (pp. 2709-2720). PMLR.
>
> [14] Tan, X., Chen, J., Liu, H., Cong, J., Zhang, C., Liu, Y., ... & Liu, T. Y. (2024). Naturalspeech: End-to-end text-to-speech synthesis with human-level quality. IEEE Transactions on Pattern Analysis and Machine Intelligence.
>
> [15] Shen, K., Ju, Z., Tan, X., Liu, Y., Leng, Y., He, L., ... & Bian, J. (2023). Naturalspeech 2: Latent diffusion models are natural and zero-shot speech and singing synthesizers. arXiv preprint arXiv:2304.09116.
>
> [16] Ju, Z., Wang, Y., Shen, K., Tan, X., Xin, D., Yang, D., ... & Zhao, S. (2024). Naturalspeech 3: Zero-shot speech synthesis with factorized codec and diffusion models. arXiv preprint arXiv:2403.03100.

---

> ### Comment · Reviewer_abUH · 2024-11-28
>
> Thank you to the authors for their response. Below are my comments.
>
> > W1 Orig. & Sign.
>
> 1. Thank you for providing a detailed explanation of the claims and rationale behind Sway Sampling. However, the reasons provided to support the novelty of the proposed method are not convincing to me for the following reasons:
>     -  It is well established that diffusion modeling allows the inference-time schedule to be modified independently of the training-time schedule (i.e., without retraining the model) [1]. Additionally, improving performance through highly interpretable solvers is not a novel concept [2]. Applying a different scheduler to E2 TTS essentially involves using an alternative existing scheduler, and the lack of performance improvement in such cases would be more surprising than its presence.
>     - This scalability stems from the inherent properties of diffusion models, not from the proposed methodology itself. Therefore, the claim that the method is "easily applied to other research fields" could not substantiate its novelty. The idea that it would perform well in other domains is a promising hypothesis that needs to be validated through rigorous experimentation in the target domain.
>     - The core idea of Sway Sampling lies in dividing diffusion time steps unevenly. However, the concept of defining and optimizing the noise schedule as a time-step optimization problem to improve performance has already been well established [3].
>     - The ability to apply various samplers, including Euler, is not a unique advantage of the proposed Sway Sampling but rather a characteristic of FM ODE-solvers, to which Sway Sampling belongs [4].
>
>     For these reasons, I feel that the proposed Sway Sampling demonstrates very limited scientific novelty, and the fact that it can improve final performance is not particularly groundbreaking (independent of the fact that the performance is better than E2 TTS). Separate from novelty, the only contribution I see is that it demonstrates, for the first time, the ability to improve the inference-time scheduler within the newly proposed TTS modeling framework of E2 TTS. However, I remain uncertain whether this finding is significant enough to be shared through ICLR, rather than a speech-specific conference, to reach a broader audience of researchers across various fields.
>
> 2. Thank you for the detailed point-to-point explanation.
>     - My intent was not to critique the distinction between AR and Non-AR models but to highlight that the authors seem unaware of the older origins of the cross-attention-based textual conditioning method, as well as to point out the lack of a related work section addressing the core issues discussed in this paper.
>     - My comment on unfamiliarity with classifications refers to the error of grouping two distinct text conditioning methods—Voicebox and E2 TTS—into the same category and misusing the term "in-context learning" to describe this category. E2 TTS represents a completely new approach, distinct from the two well-established categories (Cross-attention and Phoneme-level alignment) previously clarified in the last responses.
>     - Regarding Seed-TTS, the authors previously stated in their response ("Replying to Response to Reviewer abUH [Cont. 1/2; W1,2]" starting with "To make it clear,") that "Seed-TTS, DiTTo-TTS, and E2 TTS are concurrent works." I would like to ask why Seed-TTS, which had not been previously discussed, was suddenly mentioned in that context. Based on the subsequent discussion, I understand it to be cited alongside DiTTo-TTS as an example of models using cross-attention, but clarification on this point would be appreciated.
>
> I feel that the authors' responses, particularly the third point under item 2, deviate somewhat from the core of the discussion that should focus on the most concerned points. As a reviewer, I believe this paper falls short as a scientific paper, and I have raised critical questions that must be addressed to support its claims. To redirect the discussion appropriately, I would like to reiterate the key questions I have raised since the beginning, to refocus on the essential issues.
>
> - What exactly is the definition of "text refinement" as claimed by the authors? By what mechanism does ConvNeXt address the robustness issues of E2 TTS? Is ConvNeXt the only viable solution?
> - Why does the "entanglement of acoustic and semantic features in E2 TTS" occur? Could this be a fundamental issue stemming from the newly proposed text conditioning method in E2 TTS (combining speech and text without explicit alignment)?
> - By what mechanism does Sway Sampling address the robustness issue?
>
> Through prior discussions with the authors, **it is clear that my concerns remain unresolved, and without satisfactory answers to address them, I find no reason to change my evaluation**.

---

> > ### Comment · Reviewer_abUH · 2024-11-28
> >
> > > W2 Quality
> >
> > Thank you for the detailed explanation provided with the additional information. However, my concern regarding the quality remains unresolved, and I feel that my reasoning may not have been fully conveyed. To reiterate, my critique is not that the final objective metric scores are insufficiently high, but rather that there seems to be a discrepancy between the quality inferred from the demo samples and the objective scores reported in the paper's Table when considering the broader test set.
> >
> > 1. As the authors mentioned, the demo samples were randomly selected (no cherry-picking), and I am left to evaluate based solely on those samples.
> >     - I found that this could lead to conclusions with low reliability. Thus, considering the impracticality of listening to all samples, I requested the CI as an indicator to estimate the quality of the remaining samples.
> >     - However, the results shared by the authors show an extremely small CI (calculated over 1,127 samples, which I consider statistically reliable). This suggests that the remaining samples are of a similar quality to the demo samples, and the issues I pointed out in the demo samples during my initial review may also persist in the other evaluation samples.
> >     - The reported metrics—1.806% WER and 0.669 SIM—are undeniably high. However, if the test set consists of samples with similar quality issues to those in the demo set, such scores seem implausible. This is my current assessment.
> >     - To clarify my expectations for a resolvable scenario: If the demo samples were randomly selected but the CI was large, it would imply that the poor-quality samples happened to appear in the demo by chance (no cherry-picking), and better-quality samples could exist among the rest. In that case, the reported metrics could be trusted. Conversely, if the samples were cherry-picked but the CI was high (while acknowledging the cherry-picking), it would suggest that the remaining samples are also likely to be of high quality, allowing the final scores to be trusted. Based on the authors' responses so far, neither of these scenarios seems to apply.
> >
> > 3. Thank you for acknowledging my expertise in speech area.
> >     - In my initial review, I stated, "The paper mentions that it 'simply estimates the duration based on the ratio,' yet there appears to be a significant deviation when ground truth is provided." This comment was based on observing a substantial performance gap between cases where GT duration was provided and where it was not. Naturally, I was curious about how much this gap could be reduced.
> >     - I also mentioned, "Given the performance gap, it might be beneficial to include additional modules to improve duration prediction, even if it compromises some efficiency." As the authors noted, this suggestion was not intended to address a core RQ but rather to propose that such experiments could strengthen the paper’s scientific contribution.
> >     - Similar to my point in item 1, while discussions about the discrepancy between qualitative sample quality and absolute objective metrics may be necessary, the relative differences between objective metrics remain meaningful. If the authors provide compelling analyses to justify their results, I have no reason to doubt their validity and would gladly acknowledge their conclusions.
> >
> > I sincerely hope that my response helps the authors better understand my concerns. I would like to emphasize once again that addressing my core questions, as outlined in "W1 Orig. & Sign.," is the most critical step toward resolving these concerns.
> >
> > > References
> >
> > [1] Song, J., Meng, C., & Ermon, S. (2020). Denoising diffusion implicit models. arXiv preprint arXiv:2010.02502.
> >
> > [2] Lu, C., Zhou, Y., Bao, F., Chen, J., Li, C., & Zhu, J. (2022). Dpm-solver: A fast ode solver for diffusion probabilistic model sampling in around 10 steps. Advances in Neural Information Processing Systems, 35, 5775-5787.
> >
> > [3] Xue, S., Liu, Z., Chen, F., Zhang, S., Hu, T., Xie, E., & Li, Z. (2024). Accelerating Diffusion Sampling with Optimized Time Steps. In Proceedings of the IEEE/CVF Conference on Computer Vision and Pattern Recognition (pp. 8292-8301).
> >
> > [4] Lipman, Y., Chen, R. T., Ben-Hamu, H., Nickel, M., & Le, M. (2022). Flow matching for generative modeling. arXiv preprint arXiv:2210.02747.

---

> > > ### Author Response · Authors · 2024-11-28
> > > **Response 5 to Reviewer abUH [1/2]**
> > >
> > > Thanks for the reviewer's comments, but we think there is some basic knowledge for researching in your previous response that should be corrected. \
> > > Also, we hope the reviewer may not ignore a certain part of our discussion over and over again as you have already done for Sway Sampling twice "unintentionally". \
> > > **We are willing to make sure that our previous discussion is not just partially covered.**
> > >
> > > > W1 Orig. & Sign.
> > >
> > > 1. The reviewer has made fatal errors in our Sway Sampling strategy. We would love to address your concerns, but we hope you should be honest and not pretend to know what you don't know. We recommend that the reviewer reread our paper carefully and supplement basic mathematical knowledge before delivering comments.
> > >    1. The reviewer misclassified our Sway Sampling strategy in the first feedback and is still making the wrong definition. The Euler, midpoint and Heun ODE solvers are fixed-step, where the timesteps of flow matching are fixed based on chosen NFE steps and are used to solve the ODE with initial conditions. We need to seriously ask the reviewer, how can we solve an ODE with our Sway Sampling if you classify it as an ODE solver.
> > >    2. The reviewer has distorted the ICLR's definition of research. We are exploring the inference-time sampling strategy which is not yet fully explored compared to the training-time scheduler. Or the reviewer is implying the completeness of this field? If yes, analogously, EDM, Flow Matching, and many important works are just out of the reviewer's sight since are all diffusion models.
> > >    3. The reviewer's description of `dividing diffusion time steps unevenly` reflects a relatively shallow understanding of our Sway Sampling strategy, which explicitly biases timesteps to smaller sides. We have clearly explained our motivation and intuition in our paper and all through the discussion phase with valid experiments. We may kindly ask the reviewer in which previous work this method `has already been well established`.
> > >    4. We have also verified the scalability on E2 TTS, not only our F5-TTS. Viewing under the scope of Generative Models, with extensive experimental experiences of training CFM-based TTS models shown in this work and bearing a basic understanding of CFM mechanism (especially needed), we never doubt the feasibility and profitability of our method in other fields.
> > >    5. `By what mechanism does Sway Sampling address the robustness issue?` We would appreciate the reviewer being as rigorous as before though we totally understand because we have made a detailed discussion. Please review our **RQ3**.
> > > 2. For point 2, we have already detailed in previous response. In short,
> > >    1. We expect some referenced NAR models that have acceptable performance (which is matched with the reviewer's previous statements regarding an approach works in practice) using cross-attention without phoneme-level alignment earlier than E2 TTS (since the reviewer insists on the earliness of the former one).
> > >    2. Seed-TTS, DiTTo-TTS, and E2 TTS are listed in **chronological order**, and are concurrent works that `works in practice` under the new paradigm (no phoneme-level alignment). We have never mentioned that Seed-TTS is using cross-attention, and it is clearly an oversight or unfamiliarity of related works of the reviewer.
> > >    3. **Once again, point 2.1 is crucial. The core issues of our discussion should be based on a concrete epistemic premise.**

---

> > > ### Author Response · Authors · 2024-11-28
> > > **Response 5 to Reviewer abUH [2/2]**
> > >
> > > > W2 Quality
> > >
> > > 1. We assumed that the reviewer is requiring the CIs for your rigorosity, though it turns out to be a certain absence of statistical knowledge which truly makes us worried about the comprehensiveness of the reviewer's judgment.
> > >    1. The reviewer considered no statistical bias in your subjective assessment regarding individual samples. Although it might be an abuse of terminology, may we kindly ask what is the CI of a single person on a few dozen samples?
> > >    2. As we have mentioned in our previous response, the test sets contain samples with two orders of magnitude than your observed ones. Also, during subjective evaluations, we have invited 20 natives to test dozens of rounds, which is of much higher reliability than a single person's opinion. However, the reviewer completely reverses the causal and effect logic and solely based on the picked samples with defects to infer the overall quality, which makes us really confused.
> > >    3. We emphasize again that it was the reviewer who initially questioned the fairness and correctness of our objective metrics, which we are able to address with strong experimental results.
> > > 2. Since the reviewer acknowledges your own expertise in speech area and is eager to have a discussion focusing on scientific problems:
> > >    1. Why our consideration of possible distortion leveraging current ASR models to transcribe is intentionally ignored again, given the background that the simply estimated duration is already allowing great performance and even better than ground truth in some test sets?
> > >    2. As in our **RQ**, we follow the KISS philosophy, and a simple estimate way also meets our goal. We were also puzzled by the reviewer's flexible criteria for scientific and technique, given an unclear RQ and your neglect of existing experimental results. Specifically, in our previous response, we carefully asked the reviewer what insight you would like to see rather than just purely requiring an ablation study that `Given the performance gap, it might be beneficial to include ...`, but received no clear response, which makes us worry about your standard of `scientific contribution`.
> > >
> > >
> > > With our most sincere words put forward, we are looking forward to the reviewer's feedback **point-by-point in order to make sure the previous discussion is not just partially covered**.

---

> > > > ### Comment · Reviewer_abUH · 2024-11-29
> > > >
> > > > Thank you for your response. I understand that focusing on specific points may have led the authors to feel some areas were insufficiently addressed. In response to the request for explicit confirmation on each item, I have summarized my point-by-point answers below. I hope this addresses the authors' questions and alleviates any concerns about the coverage of the discussion.
> > > >
> > > > > W1 Orig. & Sign.
> > > >
> > > > 1. Thank you for highlighting the points of confusion and allowing me to clarify.
> > > >     - I apologize for incorrectly classifying Sway Sampling as an ODE solver. However, my primary concern remains regarding the novelty of adjusting inference-time timestep schedules to improve performance. While Sway Sampling introduces a specific bias in timestep allocation, modifying inference-time schedules has been explored in previous research, as I cited earlier.
> > > >     - My intention was not to imply that inference-time sampling strategies are fully explored or to distort ICLR's definition of research. I recognize that the work contributes to an active area of study and acknowledge the authors' contribution in confirming findings within the FM-based TTS framework. However, I believe that FM shares fundamental modeling principles with diffusion models, and they can offer similar insights.
> > > >     - I understand that Sway Sampling explicitly biases timesteps toward smaller values. Nonetheless, the idea of allocating more inference steps to certain regions has been considered in earlier studies I cited previously.
> > > >     - I appreciate the authors' efforts in demonstrating scalability within TTS. However, substantiating the claim that the method can be easily applied to other fields would require evidence or discussion beyond TTS.
> > > >     - I acknowledge that Sway Sampling focuses on sensitive regions to improve outputs. However, my concerns mentioned above still remain.
> > > >
> > > > 2. My response is as follows.
> > > >     1. Cross-attention, since its introduction in Transformer [1], has been foundational in sequence modeling and remains key in TTS for text-speech alignment. E2 TTS introduces a new conditioning method, and this paper aims to address its inherent robustness problem. Since RQ1.1 investigates the causes of this issue, it is reasonable to verify whether this is a new problem specific to E2 TTS's method. I acknowledge that models like DiTTo-TTS are concurrent works, however, it does not justify excluding comparisons with cross-attention conditioning method.
> > > >     2. I understand that the authors did not refer to Seed-TTS as cross-attention-based. The mention of Seed-TTS seemed sudden, and my intention was simply to seek clarification on its relevance. Given the authors have acknowledged my familiarity with related work, I believe any misunderstanding stems from the ambiguity in the original response rather than a lack of knowledge on my part.
> > > >     3. I fully agree with the authors' point and wish to focus our efforts solely on the core discussions.
> > > >
> > > > [1] Vaswani, A. (2017). Attention is all you need. Advances in Neural Information Processing Systems.
> > > >
> > > > > W2 Quality
> > > >
> > > > 1. I realized that I had overlooked that the provided CI referred to a single sample, and I apologize for the confusion. Thank you for pointing this out clearly. As noted, interpreting a single-sample CI as I did is statistically problematic, and I accept the critiques in points 1-3 of the authors' response. However, my original intention remains: given the impracticality of listening to all samples, I requested CI to infer the quality of other samples for fair evaluation. To move forward, could the authors provide CI for F5-TTS in Tables 1 and 2? Alternatively, if there are other methods to verify my quality assessments without listening to all samples, I would appreciate the authors' suggestions.
> > > > 2. I did not ignore the issue but did not elaborate because I didn't think it would help clarify my questions.
> > > >     - I acknowledge that ASR errors can lead to incorrect estimations and understand that the existing method outperforms GT in some cases. My original question stems from lines 417–419 of the manuscript, indicating that WER performance improves when GT is used. I was curious whether this gap could be reduced using the duration predictor and, if so, by how much.
> > > >     - The scientific contribution I envisioned, with the clear RQ—how much the gap could be reduced—could be demonstrated through a simple ablation study, including WER, SIM (or MOS) metrics for cases using GT, a duration predictor, and simple estimation. Adding variations of duration predictors, as noted in my initial review, would further clarify each method’s role in reducing the gap. I want to emphasize that this suggestion aligns with the consistent standards I have maintained. While I proposed these experiments to enhance the paper’s scientific contribution, they are not a requirement for addressing this specific question in evaluating the work.

---

> > > > > ### Comment · Reviewer_abUH · 2024-11-29
> > > > >
> > > > > > Main questions.
> > > > >
> > > > > In my previous response, I listed several key questions for which I had hoped to receive direct answers, but they remain unaddressed. If this response helps clarify the authors' concerns, I plan to focus on addressing these questions in further discussions. I have restated my questions below and kindly request the authors to, for each question: (1) specify the relevant lines in the paper if the questions can be answered, or (2) provide a brief explanation if it is considered less relevant or unnecessary to address.
> > > > >
> > > > > 1. What exactly is the definition of "text refinement"?
> > > > > 2. By what mechanism does ConvNeXt address the robustness issues of E2 TTS?
> > > > > 3. Is ConvNeXt the only viable solution? Why not Transformer, Conformer, Mamba, or even simpler architectures such as Linear Layer or MLP?
> > > > > 4. Why does the "entanglement of acoustic and semantic features in E2 TTS" occur?
> > > > > 5. Could this be a fundamental issue stemming from the newly proposed text conditioning method in E2 TTS (combining speech and text without explicit alignment)?
> > > > > 6. What makes Sway Sampling unique in resolving robustness issues in E2 TTS-like modeling?
> > > > >
> > > > > I look forward to your responses to these questions and hope they will help address my concerns. If there are still disagreements regarding the relevance of these questions, I am willing to continue the discussion and seek guidance from the Area Chair to ensure we reach a mutual understanding.

---

> > > > > > ### Author Response · Authors · 2024-12-03
> > > > > > **Official Comment by Authors to Reviewer abUH [3/4]**
> > > > > >
> > > > > > > Main questions.
> > > > > >
> > > > > > The above lengthy argument may not be interesting to the reviewer, but the premise of any research discussion should be rigorous and verifiable. Because it has been ignored, we spend a lot of energy mentioning it again and again.
> > > > > >
> > > > > > 1. A jointly learned module solely for the text modality (before making concatenation with the speech modality). Line 235-243 251-252.
> > > > > >
> > > > > > 2. This is ethically questionable to just list it in your questions without cite, though we understand this could raise your interest as it is scientific. We have first mentioned it as our future research question in `[Cont. 2/2; W3, Q2,3,5]` in response to the reviewer's request on pure ablation studies on model structures. Exploring the interpretability of neural networks is never easy, which is why we mentioned this as future RQ before and gave concrete measures that can be taken (`the analysis of how ConvNeXt is refining padded text input (e.g. analyze the output of ConvNeXt, probe E2 TTS and F5-TTS transformer first layers)`).
> > > > > >
> > > > > > 3. No, ConvNeXt is not proven to be the only viable solution. However, our efforts of making extensive experiments and offering the empirical result that a lightweight module for text to allow DiT works in practice should not be taken for granted. Otherwise, researchers would likely continue using much more complex designs. We have been emphasizing that ablation studies for pure model structure search are actually inconsistent with the scientific standards declared by the reviewer, but you have not responded positively (does adding pure ablation to prove that ConvNeXt is the preferable one critical to our proposal of a simple yet effective module?). And Q2, which we propose, is the possible direction that we believe can really explain the effectiveness (deeper mechanism than line 431-450). This alone can be regarded as a very meaningful work, which may be expected by the reviewer, but this does not constitute a reason for the reviewer to treat other contributions unfairly in our current work. `or even simpler architectures such as Linear Layer or MLP?`, this is what we could help provide some insights. In the early days of trying to reproduce E2 TTS and explore the structure of F5-TTS, we tried simple MLP, which would play a negative role.
> > > > > >
> > > > > > 4. We need to emphasize again that the reviewer should rationally follow the logic of the analysis, rather than directly based on the problem unfolding of entanglement we locate. The researchers were possibly only looking at metrics before our disclosure, and E2 TTS is better than many other models. Have you thought about its possible entanglement issue before or just puzzled about why it works well? It is never easy to find the inner problem and it already takes several pages to show that it is not fiction, though appears obvious after told. As we have mentioned and you may desire, a whole paper is worth elaborating on if continue and focus on this specific question. And we would appreciate any concrete plan or detailed suggestion following your `why` (which is recommended by the Reviewer Guide). In this paper, the related contribution we made is to take the first step, identifying, and addressing the entanglement issues with extensive and concrete experiments and with our advanced design. Continued with R-Q5.
> > > > > >
> > > > > > 5. No, we cannot say for certain that the entanglement is only with E2 TTS. As we have mentioned in R-Q4, there are many other models not as good as E2 TTS under the new paradigm. Specifically, previous works commonly explain why adding extra modules works, but rarely analyze the underlying causes of failure if without, and there could be similar issues. We think the term `priors` is more subtle. E2 TTS is of weak priors of semantic input (pure text), DiTTo-TTS is of strong priors (pretrained language model as text encoder with semantic injection to codec), phoneme-level force-alignment is very strong (rigid), and our F5-TTS's ConvNeXt for text refinement is "modest" as we interpreted (jointly learned but solely for text modality). What we want to convey is that the structure of the input receiving the text modality (the `text conditioning method` you referred) may not be the key point, but how strong the prior is may be (e.g. Simple-TTS also uses cross-attention, but its performance is not good enough to be called `an approach works in practice`). Our view is that by giving the text modality a separate modeling space, but jointly trained, it can achieve high naturalness while addressing entanglement to a large extent.
> > > > > >
> > > > > > 6. In our previous response: `Please review our RQ3.`. `RQ3: Based on the observation both in training and inference, is there a better and explainable sampling strategy?` The sampling strategy explored is for a general use to CFM-based TTS models. If the reviewer is still not clear about `the observation` we are talking about that helps with proposing our highly explainable method, you could re-read section 5.2 in our paper.

---

> > > > > ### Author Response · Authors · 2024-12-03
> > > > > **Official Comment by Authors to Reviewer abUH [1/4]**
> > > > >
> > > > > We fully respect the expression of the reviewer as an individual, but based on our attitude towards research, we need to speak up for what we believe and reject practices that seem to contradict mutual understanding for the betterment of research.
> > > > >
> > > > > > W1 Orig. & Sign.
> > > > >
> > > > > 1. **The rigor and humility should be held and judgments need to be made responsibly.** Specifically, the reviewer is unfamiliar with our Sway Sampling and even misclassified it, but comments on its lack of innovation and contribution with certainty (`I feel that the proposed Sway Sampling demonstrates very limited scientific novelty`). What allows such a confident judgment without knowing it?
> > > > >
> > > > > 2. **Please review the definition of research in the ICLR Code of Ethics**. Under your standard, only completely groundbreaking work has value, while the rest of "new or substantially improved insights" based on defined problems do not meet your criteria. Specifically, the reviewer confirmed with `I believe that FM shares fundamental modeling principles with diffusion models, and they can offer similar insights.` Here Flow matching (FM) is an analogy to our sampling method and "diffusion models" is an analogy to "inference-time sampling strategies".
> > > > >
> > > > > 3. **Would need the mathematical and practical understanding of the importance of different parameterization and weighting methods for diffusion models (above) and the mechanism of CFM sampling strategies on generation tasks (following).** The reviewer stated that `The core idea of Sway Sampling lies in dividing diffusion time steps unevenly. However, the concept of defining and optimizing the noise schedule as a time-step optimization problem to improve performance has already been well established`, and `The idea of allocating more inference steps to certain regions has been considered in earlier studies.` We seriously question the reviewer's professionalism (having just misclassified our method as an ODE solver with unusual confidence). Our proposed method is to explicitly bias timesteps to smaller sides where our motivation and intuition are clearly stated with extensive and solid experiments to support the effectiveness, explainability and applicability. We are confused about the reviewer's comments of `core idea lies in ... unevenly` and `to certain regions`. We would like to ask the reviewer, why we are not biasing to larger sides or to the middle?
> > > > >
> > > > > 4. **A falsified premise of the discussion is not appreciated.** As most recently stated by the reviewer, `Cross-attention, since its introduction in Transformer [1], has been foundational in sequence modeling and remains key in TTS for text-speech alignment.`, the reviewer is still beating around the bush and not providing any concrete reference for the subjective speculation that `Researchers in the field may be surprised that alignment is achievable without cross-attention and eager to understand why.` and `Fundamentally, it remains unclear why E2-TTS-like architecture is capable of learning some degree of alignment without cross-attention mechanisms`. We have spent a lot of effort correcting the reviewer's inexplicable preference for cross-attention where no evidence of a TTS model leveraging cross-attention to text without phoneme-level force-alignment that `works in practice` (by the reviewer) earlier than E2 TTS is provided by the reviewer.
> > > > >
> > > > > 5. **Please do not exaggerate and overinterpret our responses.** This includes, but is not limited to: presuming our unfamiliarity with related works as we requested a concrete reference for the falsified premise (the reviewer stated `I was surprised to learn that the authors were unfamiliar with this classification.`); Blaming on our `ambiguity in the original response` for the reviewer's own lack of rigor and knowledge to Seed-TTS, an important related work. The reviewer stated `Among concurrent works, Seed-TTS and DiTTo-TTS, mentioned by the authors, also utilize cross-attention.` while we have only mentioned `To make it clear, Seed-TTS, DiTTo-TTS, and E2 TTS are concurrent works. Although the reviewer is giving superiority to cross-attention, ` in response to the reviewer's falsified premise, and we have clearly put in our paper that `Seed-TTS ... though not elaborated in model details.` We are making all efforts to ensure the objectivity and rigorosity which we believe are essential for research.

---

> > > > > ### Author Response · Authors · 2024-12-03
> > > > > **Official Comment by Authors to Reviewer abUH [2/4]**
> > > > >
> > > > > > W2 Quality
> > > > >
> > > > > 1. **Lacking basic statistical knowledge and giving irrational assessment**. Although the reviewer apologize at last for the fatal errors made regarding statistics under our repeated emphasis, the obviously biased judgment remains unchanged. We have already mentioned in our paper that all scores are given as the average of three random seeds (here, we provide the CIs for Tab1,2 in order, WER ±0.11 ±0.07 ±0.09, SIM ±0.004 ±0.004 ±0.005, 32NFE). We sincerely doubt the reviewer's professionalism that this question still needs further explanations? Is the action of requiring more CIs in the behavior of improving our work at the 6th round of discussion? The reviewer keeps refusing to make full examination of our demos (`given the impracticality of listening to all samples`) and is even asking we authors ourselves to think of `other methods to verify the quality assessments without listening to all samples`.
> > > > >
> > > > > 2. We do not think adding the ablation study of introducing a duration predictor is necessary. The deviated RQ `how much the gap could be reduced` regardless of the very likely distortion of ASR transcription faced with small gaps in WER (±0.5%) and the `scientific contribution` bar is obviously inconsistent with the reviewer's attitude to other more important experiments in our work. In particular, we think the simple estimated duration fits our research topic very well, and we emphasize again that we have already done a comparative experiment with GT duration info.

---

> ### Author Response · Authors · 2024-12-03
> **Official Comment by Authors to Reviewer abUH [4/4]**
>
> In any case, we deeply and sincerely appreciate the reviewer's multiple rounds of discussions. This rebuttal process has greatly enhanced various aspects of our paper and deepened our understanding of the model, which, in our view, embodies the true purpose of ICLR's open rebuttal system.
>
> To summarize our discussion, we aim not only to address the rebuttal score and technique questions but also to provide a broader perspective on the innovation and contributions of our work.
>
> We fully respect and acknowledge the impact of the E2 TTS work, which we have appropriately credited throughout the paper. At the same time, we believe our work represents a substantial step forward based on E2 TTS. Specifically, we leverage DiT with ConvNeXt as the backbone model to improve the robustness of speech-text alignment. Additionally, we proposed Sway Sampling, which significantly enhances the inference performance of the flow matching algorithm in TTS, enabling E2 TTS/F5-TTS to achieve impressive results on in-the-wild data. Furthermore, we will open-source the model, code, and the entire training and inference pipeline. We believe these contributions have the potential to result in a profound influence on TTS modeling.
>
> This mirrors the historical progression of deep neural networks in speech recognition. Before 2011, DNNs demonstrated their effectiveness on small, clean datasets. Subsequent experiments on large-scale datasets like Switchboard and Fisher further convinced the community of deep learning's transformative potential, sparking widespread adoption and innovation in the field. We believe our work will have a similar impact by using transparent and reliable experiments to validate the effectiveness of a simple yet powerful model, facilitating new insights and approaches for TTS researchers.
>
> In light of these contributions, we believe that F5-TTS has the potential to represent a significant advancement in the TTS field.

---

### Official Review · Reviewer_mAVA · 2024-11-03

**Soundness:** 3
**Presentation:** 3
**Contribution:** 2
**Rating:** 5
**Confidence:** 5

**Summary:**

The work is an incremental engineering effort based on E2-TTS.It replaces E2-TTS's original Transformer + U-Net structure with a Diffusion Transformer. Additionally, it employs a ConvNeXt V2 to encode text sequences before passing them to the transformer. Sway Sampling strategy is introduced during model inference, which performs non-uniform sampling during inference, assigning more weight to the early stages of th steps. It can enhance both the performance and efficiency of the model.

**Strengths:**

The model performance is good and the audio sample is impressive. Experimental validation is sufficient and rigorous, which well demonstrates the effectiveness of each technique. Overall it's a good exploring exmrical engineering work of E2-TTS. The writing is clear and well-structured.

**Weaknesses:**

Although this work demonstrates impressive performance, it appears to be primarily an incremental engineering exploration of E2-TTS, rather than a novel contribution.

The major contribution lies in replacing the transformer with DiT blocks and incorporating a ConvNeXt for text encoding. Most of the model structure settings are based on empirical ablation performance (see Section 5.1).
While the ablation study shows improvements brought by these techniques, the authors do not provide a theoretical analysis of their effectiveness, which overall makes the work contribution relatively weak.

Furthermore, the discussion section does not offer insightful analysis.

**Questions:**

1. If s is getting smaller (e.g. -1, -0.9), what will be the performance of the sway sampling?

2. Although this is an empirical engineering study, I'm quite curious why ConvNeXt did not result in a performance improvement for the original E2-TTS, whereas the experiments showed that it was beneficial for DiT blocks.

3. Why Conv2audio is harmful for F5-TTS, but it is helpful for text, interms of WER?

4. Sway sampling is interesting. Have you tried to observe the mel spectrogram in different sampling stages, to verify your assumptions that "that the early flow steps are crucial for sketching the silhouette of target speech based on given prompts faithfully, the later steps focus more on formed intermediate noisy output"?

---

> ### Author Response · Authors · 2024-11-20
> **Response to Reviewer mAVA [1/2]**
>
> We sincerely appreciate your constructive feedback and spotting unclear points.
>
> > **W1**: Although this work demonstrates impressive performance, it appears to be primarily an incremental engineering exploration of E2-TTS, rather than a novel contribution.
>
> We acknowledge that we have not come up with a breakthrough of backbone model innovation like Transformer in 2017, while we believe that our work has concrete contributions under the scope of "generative model" and more than simple incremental engineering effort on E2 TTS (which is very nice work simple and effective though in few first glance seems a duplicate of Voicebox just taking out the phoneme-level force-alignment). (Off-topic) we are not actually with the use of engineering as a rigid separation word in this age, as we (probably we all) are apt to see solid work with solid steps forward rather than flighty and boastful work.
>
> With our sincere words put forward, we list the impact of our work:
> - Not only prove and make transparent the simple and effective scheme of E2 TTS (we confirmed with strong results and with E2 author's check on the correctness of implementation), which liberating the upper limit of naturalness from previous phoneme force alignment, we also achieve commendable performance with relatively small model size on in-the-wild multilingual dataset compared to English-only models trained on high-quality audiobook corpus.
> - To be specific:
>   - Addressing slow convergence and deep entanglement of acoustic and semantic features of E2 TTS (which makes sense as a solid step forward to us),
>   - trained with limited model size and uncensored data (more general usability with sentence-level duration provided zero-shot TTS paradigm),
>   - our F5-TTS allows faster training, faster inference (a key factor for generative model RTF=0.15, fastest in comparison of Table 1, no specific optimization did yet),
>   - and comparable or even better performance to leading TTS systems (one or several times the model size).
>   - Not to mention that we are releasing all code, checkpoints, and every corner of details with our promising F5-TTS showing seamless multilingual code-switching and stuff to promote the whole community, which we believe are not trivial contributions.
> - The inference-time Sway Sampling is a matter of great account which seems ignored from "major contribution". This training-free and easily applicable method greatly improves CFM-based models' performance and efficiency, which is highly desirable for generative models. Since CFM is one of the major trends, our Sway Sampling can help with other fields e.g. Video generation which also takes latency as critical.
>
> > **W2**: Lack of insightful analysis in the discussion section.
>
> We would like to phrase two of our analyses here which are revealing for us and were not provided by related TTS works as far as we know:
> - Sec.5.1 Investigation of the models’ behaviors with different input conditions to take a deeper look at the reasons behind the entanglement of acoustic and semantic features of E2 TTS (objective evaluation scores in Appendix B.2). It is clear that E2 TTS design faces the problem of deep entanglement of features (WER 9.63 to 3.48), and F5-TTS succeeds to address this (WER 4.17 to 3.22, a minor gain for more standard output when dropping audio prompt; and we are not applying Sway Sampling here to make fair comparison).
> - Sec.5.2 Sway Sampling "leak and override" experiment to provide intuitive and concrete evidence of the effectiveness of our method (will elaborate with Q4).
>
> Also, with R7LV6 Q1 and R-X8VX Q2 along with Appendix B.3 & B.4, we are willing to add more observations and analyses about our Sway Sampling strategy.
>
> An analysis of training stability on varying data scales is further included in Appendix B.7 (R7LV6 Q2).
>
> | F5-TTS small | Trained on | LibriTTS | 585 hours |  Trained on |   LJSpeech | 24 hours  |
> |---|----:|:-----:|------|-------:|:-----:|------|
> |   | **WER** | **SIM** | **UTMOS** | **WER** | **SIM** | **UTMOS** |
> | Ground Truth  | 2.23 | 0.69 | 4.09 | 2.36  | 0.72  | 4.36 |
> | 100K updates | 29.5 | 0.53 | 3.78 | 5.64  | 0.72  | 4.17 |
> | 200K updates | 4.58 | 0.59 | 4.07 | 2.93  | 0.72  | 4.18 |
> | 300K updates | 2.71 | 0.60 | 4.11 | 3.26  | 0.71  | 4.12 |
> | 400K updates | 2.44 | 0.60 | 4.11 | 3.90  | 0.70  | 4.05 |
> | 500K updates | 2.20 | 0.60 | 4.10 | 4.68  | 0.70  | 3.99 |
> | 600K updates | 2.23 | 0.59 | 4.10 | 5.25  | 0.69  | 3.93 |
> (Update 2024/11/21 UTC-12, extra evaluation with UTMOS)

---

> ### Author Response · Authors · 2024-11-20
> **Response to Reviewer mAVA [2/2]**
>
> > **Q1**: Sway Sampling performance on smaller *s* (e.g. -1, -0.9).
>
> We use $s = -1$ and first-order Euler ode solver as the default values for F5-TTS in the updated main text. From Appendix B.3 Table 5, more aggressive Sway Sampling keeps improving performance and enables using the 1st-order Euler ODE solver that brings significant speed up compared to the 2nd-order midpoint.
>
> | F5-TTS 16 NFE  | RTF | LS-PC test-clean WER | SIM | UTMOS | Seed-TTS test-en WER | SIM | UTMOS | Seed-TTS test-zh WER | SIM | UTMOS |
> |------|-----|-----:|-----|-----|------:|----|-----|------:|-----|-----|
> | s=-1, Euler        | 0.15 | 2.53 | 0.66 | 3.88 | 1.89 | 0.67 | 3.76 | 1.74 | 0.75 | 2.96 |
> | s=-1, midpoint   | 0.26 | 2.43 | 0.66 | 3.87 | 1.88 | 0.66 | 3.70 | 1.61 | 0.75 | 2.87 |
> | s=-0.8, Euler     | 0.15 | 2.82 | 0.65 | 3.73 | 2.14 | 0.65 | 3.70 | 2.28 | 0.72 | 2.74 |
> | s=-0.8, midpoint | 0.26| 2.58 | 0.65 | 3.86 | 1.86 | 0.65 | 3.68 | 1.70 | 0.73 | 2.83 |
> (Update 2024/11/21 UTC-12, extra evaluation with UTMOS)
>
> > **Q2**: Why ConvNeXt not improving E2-TTS?
>
> As E2 TTS directly takes off phoneme-level force alignment while still keeping the UNet-equipped Transformer structure, it deeply entangles acoustic and semantic features (analyzed with experiment mentioned in response to W2). We speculate that E2 TTS is leveraging the whole network where all layers are skip-connected symmetrically to learn speech-text alignment. In contrast, F5-TTS is with an adaptive layernorm DiT structure free of skip connections in E2 TTS, which is not conflictive with a jointly learned refinement module.
>
> Our work is putting forward that a modest refinement is sufficient for force-alignment-free NAR TTS rather than exploring sweepingly best DiT or UNet-Transformer variant, thus we leave some possible improvements for future works (e.g. asymmetric skip connections excluding the first few layers, a pre-learned refinement plug-and-play, etc.)
>
> > **Q3**: Why ConvNeXt harmful with audio but helpful for text in terms of WER?
>
> With a fairly neat pipeline making the most of trainable parameters used, adding Conv2audio gains slight SIM while losing much WER. It makes sense because speaking style can be extracted with such a module to facilitate voice-cloning while the time series real information is eroded (to an excessive extent from experiment results, thus we state in the main text `not ideal for scaling up`). In contrast, the padded text sequence is not force-aligned, natural to benefit from a refinement operation.
>
> > **Q4**: Have you tried to observe the mel spectrogram in different sampling stages, to verify your assumptions?
>
> Yes, we have observed the intermediate output mel spectrograms. We have also analyzed the trajectories of the whole sampling process of different samples and seeds. Due to the limited main text space, we decided to phrase with a clear and easy-to-grasp "leak and override" experiment that provides an intuitive explanation for all field researchers while no less concrete verification.
>
> From $t=0.1$ where decoded samples are still of too high SNR for humans to hear the content or distinguish the corresponding mel spectrogram, the model already grasps the general form of mel (for $t>0.25$ we are able to see and hear the consistency between generated outputs in different stages, `the later steps focus more on formed intermediate noisy output`). The more successful override with Sway Sampling emphasizes that `the early flow steps are crucial for sketching the silhouette of target speech based on given prompts faithfully`.
>
> Also, from the early training stage performance, when the model is still producing gibberish, leaking some ground truth information the same way as in the above experiment enabling intelligible output. This adds to the verification of our assumption for "later steps".

---

> ### Author Response · Authors · 2024-11-25
> **Response to Reviewer mAVA**
>
> Again, we appreciate the reviewer's valuable comments. We have carefully addressed the main concerns in detail.
>
> As the paper discussion phase is approaching an end and we truly value the discussion with the reviewer, we would be grateful to hear your feedback. We hope you would find the response satisfactory and we are always open to answering further questions.

---

> > ### Author Response · Authors · 2024-12-01
> > **Response to Reviewer mAVA (Cont.)**
> >
> > As the extended discussion phase is approaching an end, we would like to further express our sincere thanks for your valuable comments and greatly appreciate it if you could kindly provide us with your feedback.
> >
> > We have meticulously addressed your previous concerns, leaving no stone unturned in our quest to ensure your satisfaction, and we always value your time and efforts during the reviewing process.

---

> ### Comment · Reviewer_mAVA · 2024-12-02
>
> Thank you for the response.
>
> However, the response contains some more unverified assumptions (e.g. what is "modest refinement" as mentioned by Reviewer abUH) over the assumptions in the manuscript itself.
>
> Additionally, the response to W1 does not address the concerns regarding the novelty (to replace backbone, encode text with ConvNeXt, use a sway sampling). Given your remark that `E2 seems a duplicate of Voicebox,` you can understand why I view your work as merely `a simple incremental improvement on E2`.
>
> I do not deny that it could be a good technique report (actually, further experiments in other domains are necessary to validate the proposed method), some techniques lack theoretical reasoning, concerning e.g. inconsistencies of ConvNeXt on F5-TTS and E2-TTS.

---

> ### Author Response · Authors · 2024-12-02
>
> Thanks for the reviewer's comments.
>
> As we have already stated in our first response `The inference-time Sway Sampling is a matter of great account which seems ignored from "major contribution".` We would like to make the discussion if the reviewer has any concerns about the novelty of our proposed sampling strategy.
>
> Furthermore, `E2 TTS (which is very nice work simple and effective though in few first glance seems a duplicate of Voicebox just taking out the phoneme-level force-alignment)` is our response which is different from the fragment taken out of context. What we are trying to express is that more complex things are not necessarily more valuable or effective, and doing subtraction is not easy and is not of less value.
>
> The `modest refinement` is not mentioned by Reviewer abUH but `text refinement` (line235-243). The description `modest` is to make comparisons with complex or rigid text preprocessing designs (e.g. the way of DiTTo-TTS: pretrained language model as text encoder and with which jointly finetuned the codec; or the phoneme-level force-alignment). The usage of the term `refinement` is for our jointly learned module solely for the text modality before making concatenation with the speech modality.
>
> We are willing to address your concerns (e.g. which specific question in the reviewer's first comments has not been fully addressed). Looking forward to the reviewer's feedback.

---

### Note · Authors · 2025-03-02

I have read and agree with the venue's withdrawal policy on behalf of myself and my co-authors.

---

### Meta-Review · Area_Chair_CQFJ · 2024-12-19

**Metareview:**

The paper proposes to use a different embedding model and a different sampling procedure to improve a nonautoregressive synthesis model, E2 TTS.

I recommend a rejection because, despite the improvement and impressive performance, the root cause of the improvement is unclear. Other than reviewer X8VX, much of the argument among reviewers and authors is around this issue. Without identifying the root cause, the paper reads as if the motivation is missing (reviewer abUH) and the contribution is limited (reviewer mAVA and 7LV6). Lengthy discussion among the reviewers and the authors were still not enough to convince the reviewers.

I have to emphasize this is not about ablation study, as several ablation experiments are already included in the paper. it's about providing a deeper understanding of the problem. For example, the paper confirms from experiments that proposed approach improves convergence, but we do not know **why** convergence is slow to begin with and **how** convergence becomes faster once we employ the changes.

This paper has great potential and I encourage the authors to carefully consider the feedback from reviewers.

**Additional Comments On Reviewer Discussion:**

There had been an intense discussion about the merits and novelty of the work. The reviewers and authors seemed to be talking past each other. The discussion was healthy, but both parties were exhausted and became impatient. I'd suggest for future situations like this, it's best for both to propose **actionable** changes, for focus on the possible **improvements** to the paper rather than arguments that don't get agreed from both sides.

---

### Decision · Program_Chairs · 2025-01-22

Reject